# JoinGym: An Efficient Query Optimization Environment for Reinforcement Learning

## Abstract

Join order selection (JOS) is the problem of ordering join operations to minimize total query execution cost and it is the core NP-hard combinatorial optimization problem of query optimization. In this paper, we present JoinGym, a lightweight and easy-to-use query optimization environment for reinforcement learning (RL) that captures both the left-deep and bushy variants of the JOS problem. Compared to existing query optimization environments, the key advantages of JoinGym are usability and significantly higher throughput which we accomplish by simulating query executions entirely offline. Under the hood, JoinGym simulates a query plan's cost by looking up intermediate result cardinalities from a pre-computed dataset. We release a novel cardinality dataset for 3300 SQL queries based on real IMDb workloads which may be of independent interest, *e.g.*, for cardinality estimation. Finally, we extensively benchmark four RL algorithms and find that their cost distributions are heavy-tailed, which motivates future work in risk-sensitive RL. In sum, JoinGym enables users to rapidly prototype RL algorithms on realistic database problems without needing to setup and run live systems.

## 1 Introduction

Deep reinforcement learning (RL) has achieved many successes in games (Bellemare et al., 2013; Cobbe et al., 2020; Schrittwieser et al., 2020) and robotics simulators (Tassa et al., 2018; Freeman et al., 2021), which has driven most of the empirical research in RL. Beyond these settings, RL has the potential to greatly impact many other real-world domains such as congestion control (Tessler et al., 2022), job scheduling (Mao et al., 2016) and database query optimization (Marcus et al., 2019; Yang, 2022; Lim et al., 2023), which is the focus of this paper. Unfortunately most systems applications do not have realistic simulators and training occurs online in a live system. This makes RL research in these domains prohibitively expensive for most labs, which is not inclusive and slows down progress. It is thus crucial to develop realistic and efficient simulators to accelerate RL research for data systems problems. In this work, we present JoinGym, the first lightweight and easy-to-use simulator for query optimization that can simulate real-world data management problems without needing to setup live systems.

In database query optimization, join order selection (JOS; *a.k.a.* join order optimization or access path selection) is the NP-hard combinatorial optimization problem of finding the query execution plan of minimum cost. Given the ubiquity of databases, the JOS problem is very practically important and this has motivated many works in search heuristics (Selinger et al., 1979; Chandra & Harel, 1980; Vardi, 1982; Cosmadakis et al., 1988) and RL (Marcus et al., 2019; Yang et al., 2022; Marcus & Papaemmanouil, 2018; Krishnan et al., 2018). The JOS problem exhibits three key challenges: (1) long-tailed return distributions, (2) generalization in discrete combinatorial problems, (3) partial observability. These challenges are common in real systems applications but are understudied since they are not captured by the popular game and robotic simulators. With JoinGym, our aim is to provide a lightweight yet realistic simulator that can motivate methodological innovations in these three underexplored areas of RL.

The novel idea that makes JoinGym so efficient is to simulate query executions completely offline by looking up the size of intermediate tables from join sequences. These intermediate result (IR) cardinalities can be pre-computed since they are deterministic and system-agnostic. Along with JoinGym, we also release a new dataset of IR cardinalities for 3300 queries on the Internet Movie

Database (IMDb). Our query set contains 100 queries for each of the 33 templates that reflect diverse user interests about movies; this is $30\times$ larger and more diverse than the Join Order Benchmark (JOB; Leis et al., 2015).

One trade-off of our offline approach is that cumulative IR cardinality is a proxy for the online runtime metrics that end users care about, *e.g.*, query latency or resource consumption. However, it is well-established that minimizing IR cardinalities is by far the problem of query optimization that has the largest impact on execution cost (Lohman, 2014; Leis et al., 2015; Neumann & Radke, 2018; Kipf et al., 2019; Trummer et al., 2021). In particular, Lohman (2014) observed that bad cost models typically account for at most 30% degradation in runtime metrics, while high IR cardinalities can cause metrics to blow up by many orders of magnitude. Importantly, focusing on cardinality minimization affords computational advantages that we leverage: IR cardinalities are deterministic and system-agnostic so they can be pre-computed. In contrast, runtime metrics are system-dependent and can only be obtained from live query executions which are slow and costly. For example, large queries (*e.g.*, $q29\_44$, $q29\_80$ in JOINGYM) can take days to process on a commercial database, especially if the RL policy selects a sub-optimal join order, while JOINGYM can simulate thousands of queries per second on a personal laptop. Thus, by reducing query optimization to its absolute core, we can provide a realistic lightweight simulator that is practically useful for RL research.

We now briefly overview the paper. In Section 3, we mathematically formulate query optimization and recall the four variants of the JOS problem that are most often used in practice. Namely, JOINGYM supports both left-deep and bushy plans as well as enabling or disabling Cartesian products, which are common heuristics to reduce search space at the slight cost of optimality. Then in Section 4, we formulate JOS as a Partially Observable Contextual Markov Decision Process (POCMDP) and describe our design choices for the state, action, and reward. The context is partially observable because query embeddings are lossy compressions of base table contents and hence cannot fully determine the IR cardinalities. Finally, in Section 5, we perform an extensive benchmarking of standard RL algorithms on JOINGYM. We observe that return distributions are long-tailed and that generalization can be improved, which motivates further methodological RL research for problems in database query optimization and beyond.

Our main contributions are summarized as follows:

1. We provide the first lightweight simulator for query optimization that enables inexpensive and inclusive RL research on said problem.

2. We release a novel dataset containing all IR cardinalities of 3300 queries on IMDb, which is used by JOINGYM and may be of independent interest, *e.g.*, cardinality estimation.

3. We extensively benchmark RL algorithms on JOINGYM and find that existing algorithms can generalize well at the 90% quantile. However, we observe that their generalization sharply worsens due to the long-tailed cost distribution. This motivates future research to address the key challenges of query optimization: 1) long-tailed returns, 2) generalization in combinatorial optimization, 3) partial observability.

In sum, JOINGYM is an efficient and realistic environment that enables rapid prototyping of algorithms for query optimization. Our aim is to make query optimization accessible to the ML&RL communities and to accelerate methodological research in the intersection of data systems and ML&RL.

## 2 RELATED WORKS

**Environments for Query Optimization.** The query optimization environment from Park (Mao et al., 2019) require a DBMS backend (*e.g.*, PostgreSQL or Apache Calcite) and setting this up correctly can already be nontrivial. Their costs are computed by either querying the DBMS's cost model, which takes seconds per step, or online query execution time, which can take hours per step. Meanwhile, our costs are computed by looking up IR cardinalities and JOINGYM can simulate thousands of trajectories per second on a standard laptop, which is orders of magnitude faster than Park's environment. Also, Park only supports the 113 queries of JOB, while JOINGYM supports 3300 queries that we selected for diverse human searches based on templates from JOB. Lim et al. (2023) describes the architecture of DB-Gym, which like Park also runs in a real DBMS and thus can be very costly.

**Improving Query Optimization with RL.** Modern database systems typically posit a fixed data correlation between tables for cardinality estimation, which may lead to sub-optimal plans when these correlation assumptions break down. Hence, JOS is still an unsolved problem (Lohman, 2014) and a natural idea is to apply data-driven approaches to more accurately estimate cardinalities (Yang, 2022). Marcus et al. (2019); Krishnan et al. (2018); Yang et al. (2022) showed that deep RL can be competitive and improve over existing heuristics in certain settings. Marcus et al. (2021); Gunasekaran et al. (2023) showed that RL can auto-tune the parameters of the underlying DBMS and achieve superior performance. These prior works laid the foundation for the RL formulation that we adopt and improve on later in the paper. However, these prior works require setting up a DBMS and executing live queries, which as mentioned before can be slow and costly. Our goal is to make JOS accessible to RL researchers, without being slowed down by the setup and costs of real databases.

**Other Related Environments.** The main RL challenges presented by JOINGYM are (1) long-tailed returns, (2) generalization and (3) partial observability. For (1), there are no RL environments based on realistic problems with long-tailed returns to the best of our knowledge. JOINGYM can serve as the first real-world benchmark for developing risk-sensitive RL that can optimize for the tail (Chow et al., 2015; Ma et al., 2021; Wang et al., 2023). For (2), Procgen (Cobbe et al., 2020) is specifically designed for testing RL generalization by using procedural generated video games. For (3), POPGym (Morad et al., 2023) provides a suite of games and navigation environments that are partially observable. There is a lack of environments for (2) & (3) beyond games, and JOINGYM can serve as such a benchmark for combinatorial optimization and systems-inspired problems.

## 3 DATABASE QUERY OPTIMIZATION PRELIMINARIES

A database consists of $N_{\text{tables}}$ tables, $\text{DB} = \{T_1, T_2, \ldots, T_{N_{\text{tables}}}\}$, where each table $T_i$ has a set of $N_{\text{cols}}(T_i)$ columns, $\text{Cols}(T_i) = \{C_{i,1}, C_{i,2}, \ldots, C_{i,N_{\text{cols}}(T_i)}\}$. A SQL query can be abstracted into three parts $q = (I, U, J)$. First, $I = \{i_1, \ldots, i_{|I|}\} \subset [N_{\text{tables}}]$ specifies the tables needed for this query. Second, $U = \{u_i\}_{i \in I}$ is a set of unary *filter predicates* such that for each $i \in I$, a filtered table $\widetilde{T}_i = u_i(T_i)$ is produced from keeping the rows of $T_i$ that satisfy the filter predicate $u_i$. The fraction of rows in $T_i$ that satisfy $u_i$ is defined as the *selectivity*, $\text{Sel}_i = |\widetilde{T}_i|/|T_i|$. Third, $J = \{P_{i_1 i_2}\}_{i_1 \neq i_2 \in I}$ is a set of binary *join predicates* that specify which columns should have matching values between two tables.

Given two tables $R$ and $S$ and join predicates $P \subset [N_{\text{cols}}(R)] \times [N_{\text{cols}}(S)]$, define their binary join as

$$R \bowtie_P S = \{r \cup s \mid r \in R, s \in S, r_a = s_b \ \forall (a, b) \in P\}, \tag{1}$$

where $r \cup s$ means concatenating rows $r$ and $s$, and $r_a = s_b$ stipulates that the $a$-th column of $r$ matches the $b$-th column of $s$ in value. Letting $\overline{P} = \{(b, a) : (a, b) \in P\}$, we restrict $P_{i_1 i_2} = \overline{P_{i_2 i_1}}$.

The *result* of $q$ is $\widetilde{T}_{i_1} \bowtie_{P_{i_1, i_2} \cup \cdots \cup P_{i_1, i_{|I|}}} (\widetilde{T}_{i_2} \bowtie_{P_{i_2, i_3} \cup \cdots \cup P_{i_2, i_{|I|}}} \cdots (\widetilde{T}_{i_{|I|-1}} \bowtie_{P_{i_{|I|-1}, i_{|I|}}} \widetilde{T}_{i_{|I|}}))$. However, there are different ways to execute the same query as different sequences of binary joins. For example, if $q = (\{1, 2, 3, 4\}, \{u_1, u_2, u_3, u_4\}, \{P_{1,2}, P_{1,3}, \ldots\})$, two different plans to execute the same query would be $\widetilde{T}_1 \bowtie_{P_{1,2} \cup P_{1,3} \cup P_{1,4}} (\widetilde{T}_2 \bowtie_{P_{2,3} \cup P_{2,4}} (\widetilde{T}_3 \bowtie_{P_{3,4}} \widetilde{T}_4))$ and $(\widetilde{T}_1 \bowtie_{P_{1,3}} \widetilde{T}_3) \bowtie_{P_{1,2} \cup P_{1,4} \cup P_{3,2} \cup P_{3,4}} (\widetilde{T}_2 \bowtie_{P_{2,4}} \widetilde{T}_4)$. The former involves the *intermediate results* (IRs) $\text{IR}_1 = \widetilde{T}_3 \bowtie_{P_{3,4}} \widetilde{T}_4$ and $\text{IR}_2 = \widetilde{T}_2 \bowtie_{P_{2,3} \cup P_{2,4}} \text{IR}_1$, while the latter involves the IRs $\text{IR}_1 = \widetilde{T}_1 \bowtie_{P_{1,3}} \widetilde{T}_3$ and $\text{IR}_2 = \widetilde{T}_2 \bowtie_{P_{2,4}} \widetilde{T}_4$. The IRs in each of these plans can have drastically different cardinalities and the plans can therefore have drastically different run-times and computational costs (Ramakrishnan & Gehrke, 2003). The IR cardinality generally depends on the selectivity of base tables and the correlation of joined columns. JOS is the problem of finding a feasible join order that achieves the minimum total size of all IRs, *i.e.*, cardinalities.

**Left-Deep vs. Bushy Plans.** All join orders are expressible as a binary tree where the leaves are (filtered) base tables $\widetilde{T}_i$ and each internal node represents the IR from joining its two children. In the DB literature, the most important types of join plans are *bushy* and *left-deep* (Leis et al., 2015). Left-deep plans iteratively join base tables with the IR of cumulative joins so far, which is represented by a left-deep tree. Bushy plans allow for any possible binary tree; *e.g.*, joining two non-base-table IRs is allowed in bushy plans but disallowed in left-deep plans. Left-deep plans usually suffice for

| Components | Left-deep JOINGYM | Bushy JOINGYM |
|---|---|---|
| Context $x$ | Query encoding described in Section 4.2. | |
| State $s_h$ | Partial plan encoding described in Section 4.2. | |
| Action $a_h$ | Table to join, from Discrete($N_{\text{tables}}$) | Edge to join, from Discrete($\binom{N_{\text{tables}}}{2}$) |
| Reward $r_h$ | Negative step-wise regret: $r_h \propto C^\star_{\text{plan\_type}}/H - c_h$ for plan_type $\in$ {left-deep, bushy} | |
| Transition $P$ | Deterministic transition of dynamic state features, described in Section 4.2. | |
| Horizon $H$ | $|I|$ | $|I| - 1$ |

Table 1: Components of JOINGYM for deciding the join order of a single query $q = (I, U, J)$. Between the left-deep and bushy variants, the key difference is their actions (table vs edge).

fast query execution and reduce the search space of bushy plans by an exponential factor (Leis et al., 2015). However, computing the optimal plan is still NP-hard (Ibaraki & Kameda, 1984).

**Disabling Cartesian Products to Reduce Search Space.** CPs are expensive join operations where no constraints are placed on the column values, *i.e.*, the CP between $R$ and $S$ is $R \bowtie_\emptyset S = \{r \cup s \mid r \in R, s \in S\}$, which always has cardinality equal to the product of the two source tables' cardinalities. Disabling (*i.e.*, avoiding) *Cartesian products* (CPs) has been a popular heuristic to reduce search space size by trading off optimality (Ramakrishnan & Gehrke, 2003). CP is indeed the most expensive binary join, but they can actually sometimes lead to smaller cumulative costs, *e.g.*, it may be beneficial to CP two small tables before joining a large table (Vance & Maier, 1996).

## 4 JOINGYM: A DATABASE QUERY OPTIMIZATION ENVIRONMENT

We formulate JOS as a Partially Observable Contextual Markov Decision Process (POCMDP), and then describe how we encode the context and observation that is implemented by JOINGYM.

### 4.1 PARTIALLY OBSERVABLE CONTEXTUAL MDP FORMULATION

Selecting the join order naturally fits into the POCMDP framework, since it is a sequence of actions (joins) and per-step costs (IR sizes) with the goal of minimizing the cumulative cost. This can be formulated as a (partially observable) contextual MDP, consisting of context space $\mathcal{X}$, state space $\mathcal{S}$, finite action space $\mathcal{A}$, horizon $H$, transition kernels $P(s' \mid s, a)$, and contextual reward functions $r(s, a; x)$, where $s, s' \in \mathcal{S}, a \in \mathcal{A}, x \in \mathcal{X}$. First, the context $x \in \mathcal{X}$ is a fixed encoding of the query $q$. The trajectory is a join plan for this query and is generated as follows. At time $h \in [H]$, the state $s_h$ is the *partial join plan* that has been executed so far, and the action $a_h$ specifies the join operation to perform now. For bushy plans, $a_h$ can be any valid edge in the join graph; for left-deep plans, $a_h$ is simply the $h$-th table to add to the left-deep tree. Performing the join specified by $a_h$ results in an IR, whose cardinality is the cost $c_h$. We define the reward as the negative step-wise regret $r_h \propto \frac{C^\star_{\text{plan\_type}}}{H} - c_h$, where $C^\star_{\text{method}}$ is the minimum cumulative cost, for plan_type $\in$ {bushy, left-deep}. For numerical stability, we normalize our reward and clip each $c_h$ by $100 C^\star_{\text{plan\_type}}$. While $r_h$ can be either negative or positive, the cumulative reward is non-positive with zero being optimal. This procedure iterates until all queried tables are joined. For bushy plans, the horizon is the number of joins, *i.e.*, $H = |J| = |I| - 1$; for left-deep plans, the $a_1$ corresponds to staging the first table, which does not perform any joins, and so $r_1 = 0$ and $H = |J| + 1 = |I|$. Table 1 summarizes this setup.

**Remarks.** First, the set of legal actions shrinks throughout the trajectory as the policy cannot choose joins that have already been selected, *i.e.*, $\mathcal{A} = \mathcal{A}_1 \supset \mathcal{A}_2 \supset \cdots \supset \mathcal{A}_H$. We handle this by *action masking* (Huang & Ontañón, 2022), where we constrain the policy's action selection and update rules to only consider legal actions at each step.[1] Second, the transition and reward dynamics are *deterministic* and stochasticity comes from the context since queries may be sampled from some

---

[1]Another approach would be to give a large cost for choosing unavailable actions, but this would require the agent to learn to avoid such actions. We avoid this unnecessary learning with action masking.

distribution. Third, only the reward is contextual and the transition kernel is not. Our POCMDP formulation can be interpreted as a latent MDP (Kwon et al., 2021), where each query is an MDP but we do not know the full information of the query but only see a partially observable context.

**Partial Observability.** Recall that each query $q$ is encoded by a context $x$ which in particular encodes the base tables. Partial observability arises when this encoding of the base tables is lossy, *i.e.*, the encoding is not fully predictive of the IR cardinalities. Note that the Markov propery still holds for the non-contextual transition kernel and only the reward is affected by partial observability. In practical applications where base tables contain millions of rows, any tractable encoding will necessarily be lossy. The best representation for tables is still an open research question (Ortiz et al., 2018; Marcus et al., 2019; Yang, 2022). JOINGYM can be easily modified to handle new table embeddings by changing a few lines of code. Our dataset of IR cardinalities remains valid for any embedding scheme as IR cardinalities are agnostic to the table representation.

## 4.2 CONTEXT AND STATE ENCODING

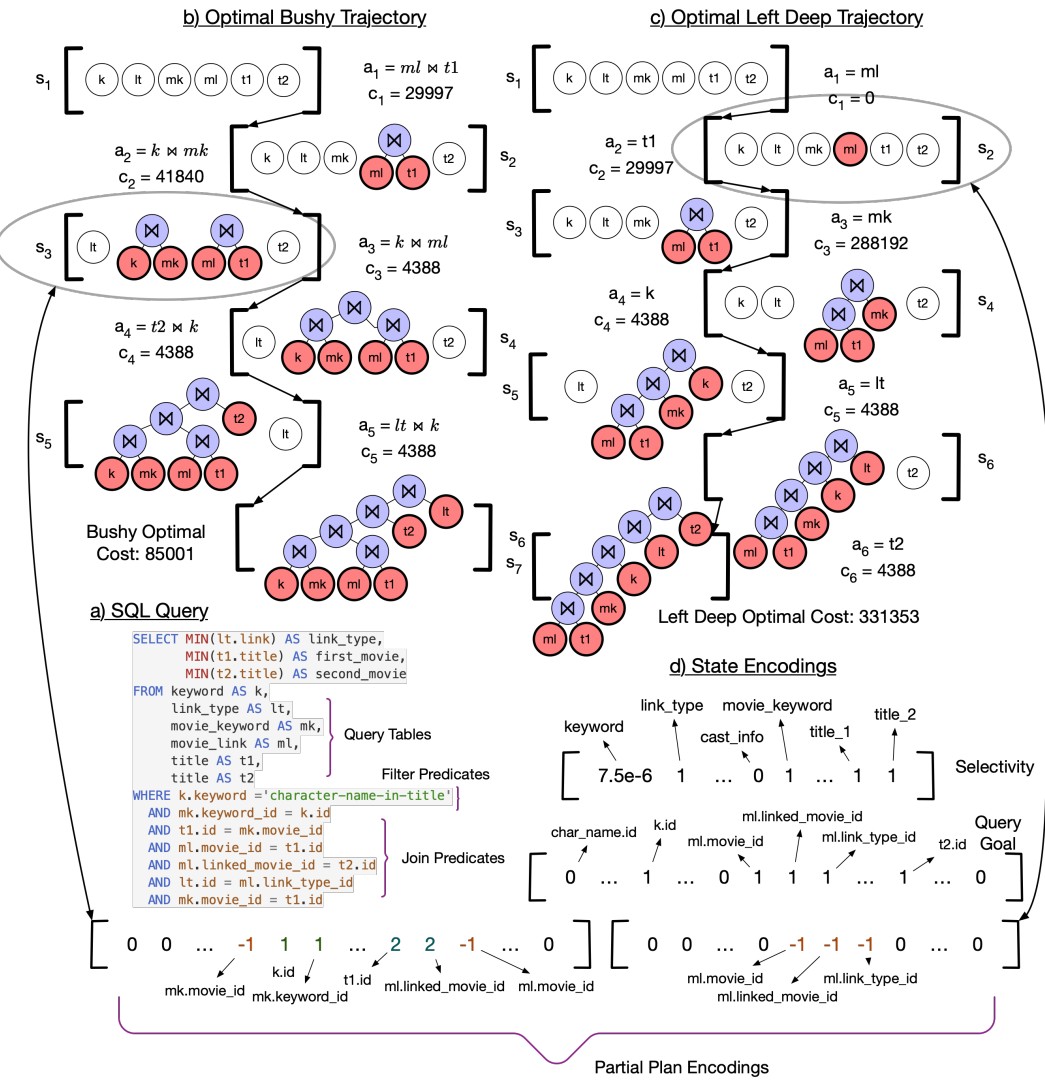

Figure 1: (a) query 110 from the JOB. (b) is its optimal bushy join plan and (c) is its optimal left-deep join plan. $c_h$ denotes the cardinality of the IR incurred at time $h$. (d) shows the query encoding (incl. selectivities) of (a) and examples partial plan encodings for left-deep and bushy states.

Given a query $q$, we represent the context $x = (v^{\text{Sel}}(q), v^{\text{goal}}(q))$ with two components that are fixed for this query. The first part is the *selectivity encoding*, a vector $v^{\text{Sel}}(q) \in [0,1]^{N_{\text{tables}}}$ where the $t$-th entry is the selectivity of $u_t$ if $t \in I$, and 0 otherwise:

$$\forall t \in [N_{\text{tables}}] : v^{\text{Sel}}(q)[t] = \begin{cases} \text{Sel}_t, & \text{if } t \in I, \\ 0, & \text{o/w} . \end{cases}$$

The second part is the *query encoding*, a binary vector $v^{\text{goal}}(q) \in \{0,1\}^{N_{\text{cols}}}$ (abusing notation, we define $N_{\text{cols}} = \sum_T N_{\text{cols}}(T)$) representing which columns need to be joined in this query; the $c$-th entry is 1 if column $c$ appeared in any join predicate, and 0 otherwise:

$$\forall c \in [N_{\text{cols}}] : v^{\text{goal}}(q)[c] = \mathbb{I}\left[\exists(R, S, P) \in J, \exists p \in P : c = p[0] \vee c = p[1]\right].$$

At time $h \in [H]$, we represent the state $s_h$ as the as *partial plan encoding* $v_h^{\text{pp}}$, which represents the joins specified by prior actions $a_{1:h-1}$. Two examples are given in Fig. 1(d). The partial plan encoding is a vector $v_h^{\text{pp}} \in \{-1, 0, 1, 2, \dots\}^{N_{\text{cols}}}$ where the $c$-th entry is positive if column $c$ has already been joined, $-1$ if the table of column $c$ has been joined or selected but column $c$ has not been joined yet, and 0 otherwise. If column $c$ has already been joined, the $c$-th entry will be the index of its join-tree in the forest; in left-deep plans, the index will always be 1 since there is always only one join-tree, but in bushy plans with more than one tree, the index can be larger than 1. It is clearly important to mark joined columns (with positive numbers) since the policy needs to know which columns have been joined to choose the next action. In addition, we also mark unjoined columns belonging to joined or selected tables with the value $-1$. For example, in left-deep plans, we must be able to tell which table was selected by $a_1$ at $h = 2$, even though said table has only been "staged" but not joined. Beyond this special case, another use-case of the $-1$-marking is that it signals marked columns as part of potentially small IR; perhaps the rows of the table has been filtered from a prior join and it is better to join with this table rather than an unjoined base table. We highlight that the partial plan encoding only logs which columns have been joined/staged rather than the current join tree, because future costs only depend on the current IRs rather than the tree structure.

### 4.3 IMPORTANCE OF GENERALIZATION IN QUERY OPTIMIZATION

In real applications such as IMDb, developers create *query templates* so that searches by a user instantiates a template with filter predicates that reflect the user's interests. A query's template determines its final query graph and different query templates may often share common subgraphs. Using the query graph structure, deep RL models can *generalize* to improve future query execution planning. For example in Fig. 2, the optimal join plan for (b) is a sub-tree of the optimal plan for (c). However, while queries with the same template have a common query graph, optimal join orders can vary significantly due to different filter conditions that are applied, *e.g.*, Fig. 2 (a) & (b) are instances of the same template (and hence share the same graph) but have different optimal join orders. Thus, the key challenge in data-driven query optimization is to learn which correct query instances to mimic based on the context (*i.e.*, filter predicates, query graph).

### 4.4 JOINGYM API

JOINGYM adheres to the standard Gymnasium API (Farama Foundation, 2023), a maintained fork of OpenAI Gym (Brockman et al., 2016). The left-deep and bushy variants are registered under the environment-ids 'joinopt_left-v0' and 'joinopt_bushy-v0' respectively. JO-INGYMcan be instantiated with `env = gym.make(env_id, disable_cp, query_ids)`, where `disable_cp` is a flag for disabling Cartesian products (described in Section 3), and `query_ids` is a set of queries that will be loaded. JOINGYM implements the abstract MDP described in Section 4.2 via the standard Gymnasium API:

(i) `state, info = env.reset(options={query_id=x})`: reset the trajectory to represent the query with id `x`, and observe the initial state. If no `query_id` is specified, then a random query is picked from the query set inputted to the constructor.

(ii) `next_state, reward, done, _, info = env.step(action)`: perform join operation specified by `action`, and observe the next state and reward. `done` is `True` if and only if all tables of the current query have been joined. There is no step truncation, so we ignore the fourth output of the Gymnasium specification for `env.step`.

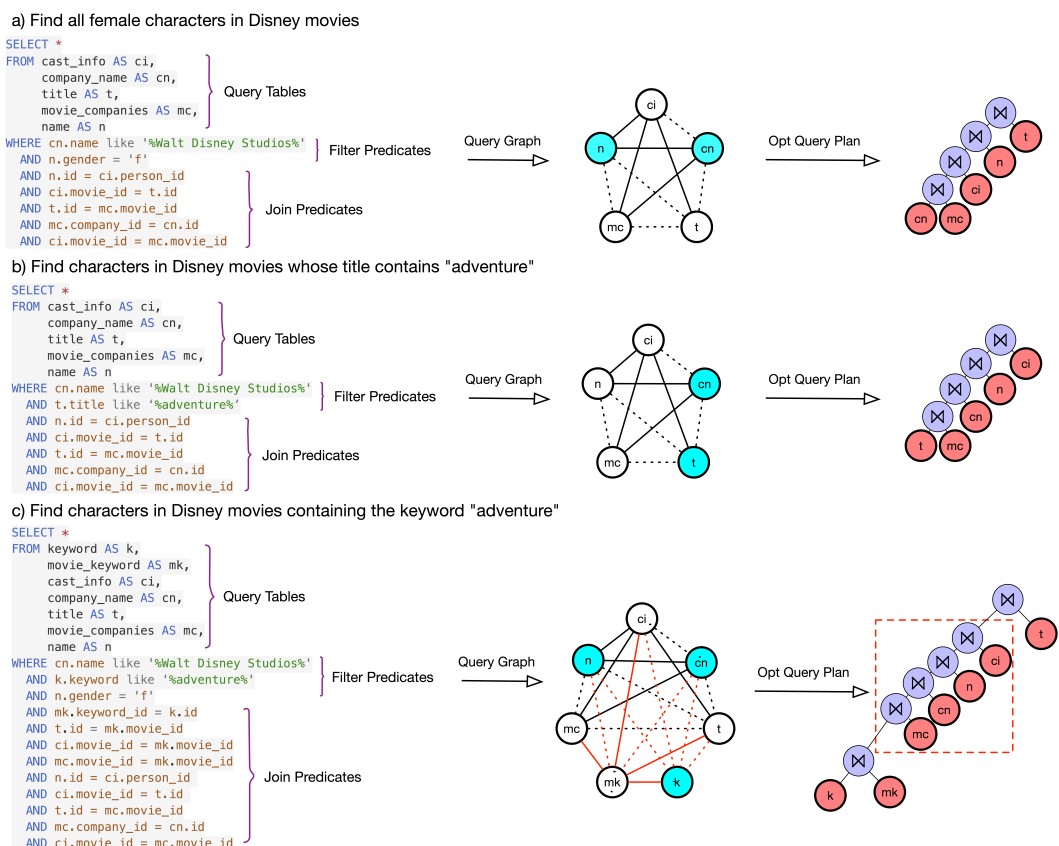

Figure 2: On the left are three raw SQL queries. The middle are their query graphs where each edge represents a join between two tables. Filtered tables are denoted by blue nodes and CPs are denoted by dashed lines. On the right, we show tree representations of the query plan, where each leaf is a table and each node is a join of its two children tables. Queries (a) and (b) are derived from the same template and so share an identical query graph but their optimal query plans are different due to different filters on the base tables. Query (c) is from a different template, but it contains (b) as a subgraph and their optimal query plans share a common sub-tree (highlighted in red).

Above, `state` & `next_state` are vector representations of the partially observed state, described in Section 4.2. Moreover, `info['action_mask']` is a multi-hot encoding of the valid actions $\mathcal{A}_h$ at the current step. The RL algorithm should take this into account, *e.g.*, masking out Q-values, so only valid actions are considered. We provide examples of how to do this in our experimental code.

## 4.5 NEW DATASET OF IR CARDINALITIES

Recall that JOINGYM contains 3300 IMDb queries, comprised of 100 queries for each of the 33 templates from the Join Order Benchmark (JOB; Leis et al., 2015). The $M$-th query from the $N$-th template has query id $qN\_M$, with $N \in [33], N \in [100]$. Our query set is $30\times$ larger and more diverse than the 113 queries of the JOB that were used in prior works (Mao et al., 2019). With JOINGYM, we release a new dataset containing all possible IR cardinalities for all 3300 queries, which was an exhaustive pre-computation we ran on hundreds of CPUs for weeks. We describe our query generation process in Appendix E. JOINGYM relies on this dataset to simulate query plan costs offline by simply looking up the IR cardinalities, instead of executing queries online. Beyond its use in JOINGYM, this dataset may also be of independent interest for research in cardinality estimation (Han et al., 2021) and representation learning for query and table embeddings (Ortiz et al., 2018).

| **Mean** | | | DQN | | DDQN | | TD3 | | SAC | | PPO |
|---|---|---|---|---|---|---|---|---|---|---|---|
| | | | RB | PER | RB | PER | RB | PER | RB | PER | |
| disable CP | bushy | trn | 8.9e+06 | 2.7e+04 | 3.6e+06 | 4.1e+04 | **1.9e+04** | 1e+05 | 8e+04 | 4.9e+04 | 1.8e+06 |
| | | val | 3.4e+04 | 1.8e+04 | 1.9e+04 | 2.4e+04 | 1.9e+04 | **1.7e+04** | 4.1e+04 | 3e+04 | 2.8e+04 |
| | | tst | 2.6e+05 | 1.4e+05 | 4.1e+05 | 8.3e+04 | 1.5e+05 | **3.4e+04** | 3.8e+04 | 3.5e+04 | 1.3e+05 |
| | left | trn | **4e+03** | 1.3e+04 | 2.2e+04 | 8.3e+03 | 1.3e+06 | 4.4e+03 | 4e+06 | 2.8e+05 | 2.1e+04 |
| | | val | 1.5e+04 | 1.2e+04 | 1.4e+04 | **9.2e+03** | 1.6e+04 | 1.3e+04 | 2e+04 | 1.2e+04 | 1.1e+04 |
| | | tst | 2.9e+05 | 1.7e+05 | 1.1e+05 | 4.5e+05 | 1.9e+04 | **1.8e+04** | 4.7e+05 | 2.5e+05 | 4.6e+04 |
| enable CP | bushy | trn | 2e+45 | 1.5e+50 | 3e+37 | 5.8e+33 | 7.8e+26 | 1.1e+24 | 1.7e+47 | 2.1e+53 | **1.1e+15** |
| | | val | 1.4e+30 | 1.6e+29 | 8.1e+25 | 5.3e+21 | 2.1e+05 | **1.8e+05** | 1.6e+42 | 3.5e+42 | 3.8e+05 |
| | | tst | 2.4e+46 | 1.7e+45 | 2.2e+41 | 8.5e+30 | 3.2e+25 | **3.6e+19** | 4e+49 | 4.1e+51 | 1.3e+28 |
| | left | trn | 3.3e+24 | 9.3e+08 | 1.9e+08 | 1.5e+14 | 1.5e+14 | 1.6e+05 | **1.3e+04** | 1.8e+04 | 7.9e+10 |
| | | val | 2.4e+04 | 1.8e+04 | 2.6e+04 | 2.2e+04 | 1.6e+05 | 2.3e+04 | **1.2e+04** | **1.2e+04** | 4.8e+04 |
| | | tst | 7.7e+14 | 1.9e+11 | 8.1e+17 | 1.4e+27 | 2.1e+14 | 1.7e+25 | **5.6e+05** | 1.1e+06 | 5.2e+23 |
| **90% Quantile** | | | DQN | | DDQN | | TD3 | | SAC | | PPO |
| | | | RB | PER | RB | PER | RB | PER | RB | PER | |
| disable CP | bushy | trn | 7.3 | **4.4** | 5.2 | 5.6 | 5.3 | 7.3 | 13 | 9.5 | 6 |
| | | val | 16 | **13** | 14 | 15 | 15 | 18 | 18 | **13** | 23 |
| | | tst | 46 | **25** | 30 | 30 | 26 | 40 | 55 | 33 | 42 |
| | left | trn | 5.5 | 5.6 | 7 | 6.5 | 6.9 | **5.2** | 11 | 8.6 | **5.2** |
| | | val | 12 | 15 | 13 | 14 | 13 | **9.5** | 20 | 14 | 11 |
| | | tst | 28 | 30 | 34 | 34 | 22 | 20 | 39 | 32 | **19** |
| enable CP | bushy | trn | 6.4e+04 | 2.4e+05 | 4.6e+04 | 3.2e+04 | 1.8e+02 | 42 | 7.7e+18 | 3e+14 | **35** |
| | | val | 2e+05 | 1.1e+06 | 5.8e+04 | 6e+04 | 3.1e+02 | 1.4e+02 | 4e+18 | 2.8e+14 | **1e+02** |
| | | tst | 1.6e+05 | 1.2e+05 | 2.1e+05 | 6.9e+04 | 2e+03 | 4.9e+02 | 2.2e+17 | 2.1e+17 | **2.8e+02** |
| | left | trn | 17 | **6.3** | 17 | 9.9 | 3.6e+02 | 17 | 7.7 | 6.8 | 9.9 |
| | | val | 25 | 15 | 27 | 22 | 2e+02 | 24 | 13 | **12** | 27 |
| | | tst | 64 | 36 | 66 | 59 | 1.7e+03 | 1e+02 | **28** | 30 | 92 |

Table 2: The top table contains the average CCM (lower is better) over the training (trn), validation (val) or testing (tst) query sets and over four possible environments. The bottom table contains the 90% quantile CCM. RB stands for "replay buffer"; PER stands for "prioritized experience replay". We highlight the best performing algorithm (lowest CCM) in each row.

## 5 BENCHMARKING REINFORCEMENT LEARNING ON JOINGYM

We present benchmarking results on JOINGYM that measure the performance of popular RL methods on the JOS task. We're specifically interested in testing the generalization abilities of these algorithms on tasks with long-tailed returns as well as partial observability.

**Experiment Setup.** Recall that our new dataset contains 100 queries for each of the 33 templates from the JOB (Leis et al., 2015). For each template, we randomly selected $60, 20, 20$ queries for training (1980 queries), validation (660 queries) and testing (660 queries) respectively. We benchmarked four different RL algorithms: (i) an off-policy Q-learning algorithm Deep Q-Network (DQN) (Mnih et al., 2015); (ii-iii) two off-policy actor-critic algorithms, Twin Delayed Deep Deterministic policy gradient (TD3) (Fujimoto et al., 2018) and Soft Actor-Critic (SAC) (Haarnoja et al., 2018); and (iv) an on-policy actor-critic algorithm Proximal Policy Optimization (PPO) (Schulman et al., 2017). For DQN, we conducted an ablation with the Double Q-learning (Van Hasselt et al., 2016). For (i-iii), we conducted ablations with standard replay buffer (RB) vs. prioritized experience replay (PER) (Schaul et al., 2015).

**Training and Evaluation.** All algorithms were trained for *one million steps* on the training queries and the best performing algorithm was selected by the cumulative cost multiple (CCM) averaged over the validation queries. We define the *cumulative cost multiple* (CCM) as the cumulative IR cardinality of the join plan divided by the smallest possible cumulative IR cardinality for this query. This can be interpreted as a *multiplicative regret* and lower is better. We swept over multiple learning rates per algorithm and also used the average CCM on the validation queries for selecting the best hyperparameter. For these best hyperparameters, we report the average CCM over the test queries in Table 2. As shown in Fig. 3, the CCM distributions are long-tailed so we also report the $p90$ CCM (90% quantile) over each query set. Our results are averaged over 10 seeds and we report the standard errors along additional $p95$ & $p99$ results in Appendix G. An example learning curve of the best algorithm (TD3) for left-deep plans with CPs disabled is shown in Fig. 4.

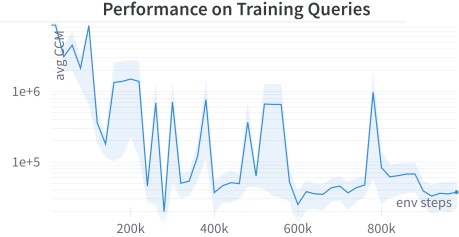
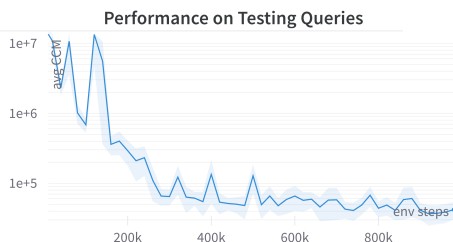

Figure 4: Learning curves of the best algorithm (TD3) for "disable CP & left-deep", where y-axis is the mean CCM and x-axis is the number of update steps. Shaded region is stderr over 10 runs.

**Discussion.** **1)** Algorithms uniformly perform much better in left-deep JOINGYM than bushy JOINGYM because the search space is exponentially smaller. For the same reason, algorithms uniformly perform better when CPs are disallowed. The hardest setting is bushy with CPs, where most algorithms diverge and those that converge have CCMs orders of magnitude worse than the other settings. **2)** Regarding generalization, the gap between the validation and test performance is only $2\times$ for $p90$ and it increases to $10\times$ for $p95$. For $p99$, both the validation and test performance are significantly worse than training. This exponential widening of the generalization gap is likely explained by the fact that the long tail is exactly where partial observability is more pronounced, causing catastrophic failure in planning of the policy. **3)** Off-policy

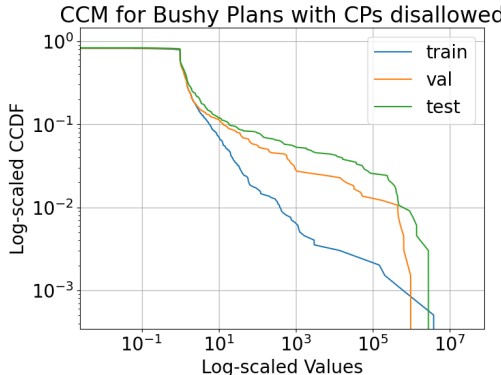

Figure 3: Complementary CDF of the CCM distribution over train, validation and test queries, of the best algorithm (TD3) for "disable CP & left-deep".

actor critic methods (TD3 & SAC) achieve the best results for the mean and are also near-optimal for the quantiles. They are more sample efficient than PPO due to their sample reuse and more stable than DQN for the query optimization task. We also find that prioritized replay does not always improve performance.

## 6    CONCLUSION

In this work, we presented JOINGYM, the first efficient simulator for query optimization based on IR cardinalities which we hope can accelerate research in the intersection of RL and systems. *For the RL community*, JOINGYM is a realistic environment for prototyping data-driven combinatorial optimization algorithms. Notably, it exhibits specific challenges in long-tailed returns, generalization and partial observability, which are underexplored by existing simulators, *e.g.*, Atari and MuJoCo. *For the systems community*, this work also provides a new dataset of IR cardinalities, which can be useful for improving cardinality estimation algorithms (Yang, 2022; Han et al., 2021) and representation learning for query and table embeddings (Ortiz et al., 2018). Finally, we believe that risk-sensitive RL can play an important role in dealing with the long-tailed returns, which are common in systems applications. In our experiments, we found that standard RL algorithms can generalize well at the 90% quantile, but their performance sharply drops at higher quantiles, *e.g.*, 95% or 99%. Risk-sensitive RL can optimize for the worst $\tau$-percent of outcomes, *e.g.*, Conditional Value-at-Risk (CVaR), which would ensure better tail performance. Existing works in CVaR RL (Lim & Malik, 2022; Urpí et al., 2021; Ma et al., 2021) have focused on noisy versions of games and MuJoCo, so it would be promising future work to develop and test such algorithms for real systems applications.

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

# Appendices

## A    LIST OF ABBREVIATIONS AND NOTATIONS

Table 3: List of Abbreviations

| | |
|---|---|
| JOS | Join Order Selection |
| DB | Database |
| IR | Intermediate result table |
| CP | Cartesian product join |
| JOB | Join Order Benchmark (Leis et al., 2015) |
| CCM | Cumulative Cost Multiple |

Table 4: List of Notations

| | |
|---|---|
| $N_{\text{tables}}$ | Number of tables in the DB |
| $N_{\text{cols}}(T)$ | Number of columns in table $T$ |
| $N_{\text{cols}}$ | Total number of columns amongst all tables in DB |
| $A \bowtie_P B$ | Binary join operator, defined in Eq. (1) |

## B    STATISTICS ABOUT JOINGYM

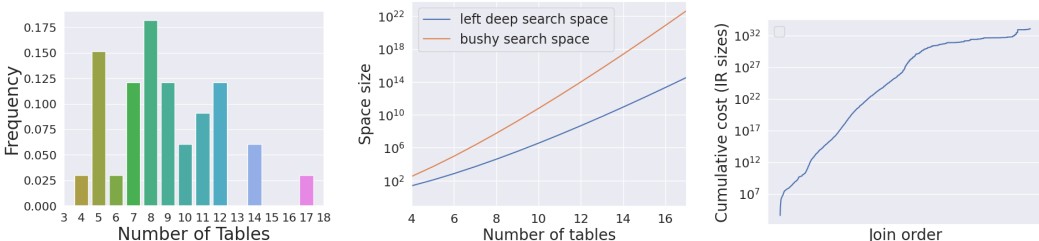

Figure 5: Left: distribution of the number of tables in JOINGYM. Middle: size of search space in JOINGYM. Right: cumulative cost (IRs) of different join orders for query q23_33.

**Search space of Left-deep vs. Bushy plans.**    Recall that left-deep plans only allow for left-deep join trees, while bushy plans allow for arbitrary binary trees. We can compute the size of the search space for both types of plans, which is a simple exercise in combinatorics. With $|I|$ tables, there are $|I|!$ possible left-deep plans and $|I|!C_{|I|-1}$ bushy plans, where the $k$-th Catalan number $C_k$ is the number of unlabeled binary trees with $k+1$ leafs. By Stirling's approximation, $n! \approx \Theta(\sqrt{n}(\frac{n}{e})^n)$ and $n!C_{n-1} \approx \Theta(n^{-1}(\frac{4n}{e})^n)$. The middle of Fig. 5 illustrates the exponential growth of two different search spaces (*i.e.*, left-deep search space and bushy search space) as the number of join tables increases. The bushy search space exhibits even faster growth compared to the left-deep plans. In our most challenging query template, search space exceeds $10^{15}$ for left-deep plans and surpasses $10^{23}$ for bushy plans. While left-deep plans usually suffice for fast query execution, bushy plans can sometimes yield join plans with smaller IR cardinalities and faster runtime (Leis et al., 2015).

**Statistics of queries in JOINGYM.**    The left of Fig. 5 shows the distribution of the number of join tables in JOINGYM. More than half of the queries join at least nine tables. The right figure shows the sorted cumulative IR sizes of different join orders for the same representative query. The left-most point is the optimal plan. The sharp jump in cardinalities from the optimal illustrates the key challenge of query optimization.

## C  FURTHER COMPARISON WITH RELATED DATABASE ENVIRONMENTS

Unlike other data system gym environments (Mao et al., 2019; Lim et al., 2023) or reinforcement learning (RL) on real database systems (Marcus et al., 2019; Yang et al., 2022; Krishnan et al., 2018), which execute queries online and use runtime costs as rewards, our environment is lightweight (i.e., it uses offline traces) and focuses on the fundamental challenge in query optimization: cardinality estimation. With the JoinGym environment, users can simulate millions of large joins within just a few minutes, whereas it takes a few hours (e.g., q24_20, q24_98), even a few days (e.g., q29_44, q29_80) to run some queries with the default configuration in PostgreSQL, which makes the other system slow and expensive.

**Why IR cardinality is a good proxy for the cost of a query plan?**

1. Lohman (2014) puts it eloquently, "The root of all evil, the Achilles Heel of query optimization, is the estimation of the size of intermediate results, known as cardinalities. In my experience, the cost model may introduce errors of at most 30% for a given cardinality, but the cardinality model can quite easily introduce errors of many orders of magnitude! Let's attack problems that really matter, those that account for optimizer disasters, and stop polishing the round ball."

2. Neumann & Radke (2018) adopts IR cardinalities as the key metric for benchmarking query optimization algorithms. Numerous database papers focus on enhancing cardinality estimation with sketching/statistical methods (Kipf et al., 2019) and neural models (Kipf et al., 2018). Also, many database theory works (Atserias et al., 2013; Ngo et al., 2018) focus on designing new algorithms to minimize the size of the intermediate result.

3. Furthermore, the IR cardinality metric provides computational advantages, which we leverage: (i) IR cardinality does not depend on specific system configurations (e.g., IO and CPU); (ii) IR cardinality is deterministic so we can pre-compute it for all our queries; (iii) with our pre-computed dataset, users can simulate millions of large joins within just a few minutes.

In sum, IR cardinality is the most common and important metric for query optimization algorithms. At the same time, its system-independent and deterministic nature allows us to design a realistic environment that is lightweight, enabling ML & RL researchers from diverse communities to collaboratively tackle the core problem in query optimization.

**State and context embeddings.**  We use the same selectivity encoding as BALSA (Yang et al., 2022) and NEO (Marcus et al., 2019). However, their query encoding is an adjacency matrix (at the table level) that preserves the tree structure, while we encode queries with a multi-hot vector (at the column level) marking which columns should be joined, similar to DP (Krishnan et al., 2018) and REJOIN (Marcus & Papaemmanouil, 2018). To the best of our knowledge, marking the tree's component index in the partial plan encoding is novel and allows us to handle bushy plans without keeping track of the whole tree; except DP, the aforementioned works only consider left-deep trees. Note that since the graph structure does not influence future IR sizes but column information does, our encoding is more compact than prior encoding schemes of BALSA and NEO. JOINGYM is designed so that one can easily change the state and context encoding schemes without needing to collect any more data, which is the costly step of building JOINGYM that we have already finished.

## D  IMPLEMENTATION DETAILS OF JOINGYM

We now describe the specific implementation details of JOINGYM. This section is intended for advanced users who want to change how we encode state, actions or rewards. We appreciate any questions or feedback and welcome pull requests.

Our code registers two `gymnasium.Env` classes that implement bushy and left-deep join plans:

1. `JoinOptEnvBushy` (in file `join_optimization/envs/join_opt_env_bushy.py`),
2. `JoinOptEnvLeft` (in file `join_optimization/envs/join_opt_env_left.py`).

As mentioned in Table 1, the main difference between these two environments lies in their action space; a bushy plan's actions are pairs of tables, while a left-deep plan's

actions are single tables. Since their state representations are nearly identical, both `JoinOptEnvBushy` and `JoinOptEnvLeft` subclass a base class called `JoinOptEnvBase` (in file `join_optimization/envs/join_opt_env_base.py`), which we describe first.

**JoinOptEnvBase** This base class handles most of the `__init__` initialization work of loading in the database schema, loading in the IR cardinality dataset, as well as constructing the selectivity encoding $v^{\text{Sel}}(q)$ and goal encodings $v^{\text{goal}}(q)$ (defined in Section 4.2) for all the queries $q$ in our dataset. Recall that $v^{\text{Sel}}(q)$ and $v^{\text{goal}}(q)$ are static during the trajectory, so we can pre-compute them when initializing the environment.

`JoinOptEnvBase` also contains a helper function `log_cardinality_to_reward` that converts log IR cardinality at step $h$, *i.e.*, $\log c_h$, to this step's reward $r_h = \frac{1}{C_{\max}(q)}(C_{\min}(q) - \exp(\min\{\log c_h, \log C_{\max}(q)\}))$, where $C_{\max}(q) = 100 \cdot C^\star(q)$, $C^\star(q)$ is the optimal (minimum-possible) *cumulative* IR cardinality for query $q$, and $C_{\min}(q) = C^\star(q)/\text{num tables to join in } q$. To interpret this expression, note that $\exp(\min\{\log c_h, \log C_{\max}(q)\}) = \min\{c_h, C_{\max}(q)\}$. We perform the clipping since IR cardinalities can get large, especially with Cartesian products enabled; this is also why we perform clipping inside the $\exp$ and work in log-space. Next, we can interpret $\sum_h c_h - C_{\min}(q)$ as essentially the regret of this trajectory, as $C_{\min}(q) \cdot H = C^\star(q)$. Finally, the scaling by $1/C_{\max}(q)$ is for normalization. In essence, our reward is the per-step negative regret.

**JoinOptEnvLeft and JoinOptEnvBushy** Each class has three main jobs: 1) maintaining the left-deep join tree, 2) a function to compute the partial plan encoding, 3) a function for computing the valid action masks. As for (1), since left-deep and bushy trees have different structures, we maintain them in different ways, though they both use the `TreeNode` data structure to do so. For (2), the partial plan encoding can be computed by examining the `TreeNode` so far, and only retaining the useful information. Finally, since the action spaces are different, each class has different functions for the valid action mask (3). It is worth highlighting that each class has two functions for computing the valid action mask: `self.valid_action_mask()` is used when Cartesian products are allowed, and `self.valid_action_mask_with_heurstic()` is used otherwise.

## E  MECHANISM FOR GENERATING QUERIES

We use the 33 predefined query templates of the Join Order Benchmark (JOB) (Leis et al., 2015) and introduce variations in unary predicates, *i.e.*, filter statements, to generate new queries. To randomly generate realistic unary predicates, we begin by conducting a manual examination of all columns within each table to select a subset of columns that are typically used in real user queries. The columns we identified were `aka_name(name)`, `aka_title(title)`, `char_name(name)`, `comp_cast_type(kind)`, `company_name(name, country_code)`, `company_type(kind)`, `info_type(info)`, `keyword(keyword)`, `kind_type(kind)`, `link_type(link)`, `movie_companies(note)`, `movie_info(info)`, `movie_info_idx(info)`, `name(name)`, `person_info(note)`, `role_type(role)`, `title(title, production_year)`. To simulate searches by real IMDb users, we compiled the top 100 most common values for each column using ChatGPT. If any column has less than 100 unique values, we do not need to use ChatGPT and simply used all the possible values.

For example, consider the SQL template `q1`, reproduced below.

```
SELECT MIN(mc.note) AS production_note,
       MIN(t.title) AS movie_title,
       MIN(t.production_year) AS movie_year
FROM company_type AS ct,
     info_type AS it,
     movie_companies AS mc,
     movie_info_idx AS mi_idx,
     title AS t
WHERE ct.id = mc.company_type_id
  AND t.id = mc.movie_id
```

```
    AND t.id = mi_idx.movie_id
    AND mc.movie_id = mi_idx.movie_id
    AND it.id = mi_idx.info_type_id
```

We consider unary predicates from those candidate columns `company_type(kind)`, `info_type(info)`, `movie_companies(note)`, `movie_info_idx(info)`, `title(title, production_year)`. For each candidate column, we flip a coin and decide to add a unary predicate with probability $50\%$. Suppose that the coin flips for each column were respectively $1, 0, 0, 0$, so we only choose the `company_type(kind)` column to create a unary predicate. Subsequently, we pick random number $n \sim \mathrm{Unif}(\{1, 2, 3, 4, 5\})$ and take $n$ random elements from the 'top-100 list' described above. Suppose that $n = 2$ and we randomly sampled 'production companies' and 'special effects companies' from the 'top-100 list' for the `company_type(kind)` column. This process leads us to the resulting query `q1_0`.

```
SELECT MIN(mc.note) AS production_note,
    MIN(t.title) AS movie_title,
    MIN(t.production_year) AS movie_year
FROM company_type AS ct,
    info_type AS it,
    movie_companies AS mc,
    movie_info_idx AS mi_idx,
    title AS t
WHERE ct.id = mc.company_type_id
    AND t.id = mc.movie_id
    AND t.id = mi_idx.movie_id
    AND mc.movie_id = mi_idx.movie_id
    AND it.id = mi_idx.info_type_id
    AND ct.kind in ('production_companies',
        'special_effects_companies')
```

Note the key addition is the last filter statement, `AND ct.kind in ('production companies', 'special effects companies')`. We repeat this procedure 99 more times to produce `q1_0, ..., q1_99`. We repeat the above for the 32 remaining templates `q2, ..., q33`, which yields the $100 \times 33 = 3300$ random queries that make up our new dataset.

## F  LIMITATIONS

**Multiple base tables in a query.**   Our current solution is to introduce duplicate tables and treat tables from the same basetables differently. Given query templates, our encoding has $n$ positions for a basetable, where $n$ is the maximum number of times this basetable appears among all query templates. We assume that query templates are fixed. We acknowledge that this solution may not be elegant and can be improved in future work.

**Dynamic workload.**   In this benchmark, we assume the RL agent is trained and evaluated on the same database, *i.e.*, we assume the DB content is kept static as in prior works Yang et al. (2022). However, in real applications, the database may dynamically change over time. It is possible to add more queries and databases to JOINGYM by simply running our data collection script to collect more cardinality data.

## G  ADDITIONAL RESULTS FOR ONLINE RL

The following tables show the mean, $p90$, $p95$, $p99$ results with standard error confidence intervals computed over 10 seeds.

| **Mean** | | | DQN | | DDQN | | TD3 | | SAC | | PPO |
|---|---|---|---|---|---|---|---|---|---|---|---|
| | | | RB | PER | RB | PER | RB | PER | RB | PER | |
| disable CP | bushy | trn | 8.9e+06 (8.8e+06) | 2.7e+04 (2.6e+04) | 3.6e+06 (3.6e+06) | 4.1e+04 (3.7e+04) | **1.9e+04 (1.1e+04)** | 1e+05 (5.7e+04) | 8e+04 (3.2e+04) | 4.9e+04 (1.3e+04) | 1.8e+06 (1.7e+06) |
| | | val | 3.4e+04 (1e+04) | 1.8e+04 (2.5e+03) | 1.9e+04 (3.3e+03) | 2.4e+04 (2.8e+03) | 1.9e+04 (3.1e+03) | **1.7e+04 (3.7e+03)** | 4.1e+04 (4.1e+03) | 3e+04 (7.5e+03) | 2.8e+04 (2.5e+03) |
| | | tst | 2.6e+05 (1.4e+05) | 1.4e+05 (3.3e+04) | 4.1e+05 (1.4e+05) | 8.3e+04 (2.5e+04) | 1.5e+05 (9.2e+04) | **3.4e+04 (6e+03)** | 3.8e+04 (1e+04) | 3.5e+04 (9.8e+03) | 1.3e+05 (4.6e+04) |
| | left | trn | **4e+03 (7.7e+02)** | 1.3e+04 (6.9e+03) | 2.2e+04 (1.1e+04) | 8.3e+03 (5.8e+03) | 1.3e+06 (1.3e+06) | 4.4e+03 (2.7e+02) | 4e+06 (1.7e+06) | 2.8e+05 (2.7e+05) | 2.1e+04 (1e+04) |
| | | val | 1.5e+04 (1.5e+03) | 1.2e+04 (1.3e+03) | 1.4e+04 (2.6e+03) | **9.2e+03 (1.2e+03)** | 1.6e+04 (3.7e+03) | 1.3e+04 (1.4e+03) | 2e+04 (2.4e+03) | 1.2e+04 (1.4e+03) | 1.1e+04 (1e+03) |
| | | tst | 2.9e+05 (2.3e+05) | 1.7e+05 (8.8e+04) | 1.1e+05 (6.2e+04) | 4.5e+05 (2e+05) | 1.9e+04 (3.7e+03) | **1.8e+04 (4e+03)** | 4.7e+05 (1.9e+05) | 2.5e+05 (1.1e+05) | 4.6e+04 (2.6e+04) |
| enable CP | bushy | trn | 2e+45 (2e+45) | 1.5e+50 (1.5e+50) | 3e+37 (2.9e+37) | 5.8e+33 (5.7e+33) | 7.8e+26 (7.8e+26) | 1.1e+24 (1.1e+24) | 1.7e+47 (1.7e+47) | 2.1e+53 (2.1e+53) | **1.1e+15 (1.1e+15)** |
| | | val | 1.4e+30 (1.4e+30) | 1.6e+29 (1.6e+29) | 8.1e+25 (5.4e+25) | 5.3e+21 (5.3e+21) | 2.1e+05 (6.8e+04) | **1.8e+05 (4.2e+04)** | 1.6e+42 (8.8e+41) | 3.5e+42 (2.2e+42) | 3.8e+05 (1.1e+05) |
| | | tst | 2.4e+46 (2.4e+46) | 1.7e+45 (1.7e+45) | 2.2e+41 (2.2e+41) | 8.5e+30 (7.5e+30) | 3.2e+25 (3.2e+25) | **3.6e+19 (3.6e+19)** | 4e+49 (4e+49) | 4.1e+51 (3.4e+51) | 1.3e+28 (1.3e+28) |
| | left | trn | 3.3e+24 (3.3e+24) | 9.3e+08 (9.3e+08) | 1.9e+08 (1.4e+08) | 1.5e+14 (1.5e+14) | 1.5e+14 (1.5e+14) | 1.6e+05 (7.1e+04) | **1.3e+04 (7.3e+03)** | 1.8e+04 (1.1e+04) | 7.9e+10 (5.6e+10) |
| | | val | 2.4e+04 (5e+03) | 1.8e+04 (2.9e+03) | 2.6e+04 (3.1e+03) | 2.2e+04 (2.1e+03) | 1.6e+05 (5.7e+04) | 2.3e+04 (4.1e+03) | **1.2e+04 (1.3e+03)** | **1.2e+04 (8e+02)** | 4.8e+04 (1.2e+04) |
| | | tst | 7.7e+14 (5.5e+14) | 1.9e+11 (1.5e+11) | 8.1e+17 (7.5e+17) | 1.4e+27 (1.4e+27) | 2.1e+10 (2.1e+10) | 1.7e+25 (1.7e+25) | **5.6e+05 (1.9e+05)** | 1.1e+06 (7.5e+05) | 5.2e+23 (5.2e+23) |

| 90% **Quantile** | | | DQN | | DDQN | | TD3 | | SAC | | PPO |
|---|---|---|---|---|---|---|---|---|---|---|---|
| | | | RB | PER | RB | PER | RB | PER | RB | PER | |
| disable CP | bushy | trn | 7.3 (2.3) | **4.4 (0.11)** | 5.2 (0.17) | 5.6 (0.52) | 5.3 (0.32) | 7.3 (0.74) | 13 (1.3) | 9.5 (0.73) | 6 (0.32) |
| | | val | 16 (2.7) | **13 (0.71)** | 14 (0.85) | 15 (0.75) | 15 (1.5) | 18 (1.9) | 18 (1.8) | **13 (0.8)** | 23 (2) |
| | | tst | 46 (15) | **25 (2.5)** | 30 (5.4) | 30 (2.9) | 26 (3.6) | 40 (7.7) | 55 (11) | 33 (3.1) | 42 (4) |
| | left | trn | 5.5 (0.51) | 5.6 (0.37) | 7 (1.1) | 6.5 (1.4) | 6.9 (0.98) | **5.2 (0.43)** | 11 (0.99) | 8.6 (0.69) | **5.2 (0.92)** |
| | | val | 12 (0.73) | 15 (1.7) | 13 (0.79) | 14 (2) | 13 (1.7) | **9.5 (0.41)** | 20 (1.5) | 14 (1.1) | 11 (1.5) |
| | | tst | 28 (3.1) | 30 (2.6) | 34 (3.7) | 34 (3.2) | 22 (3.1) | 20 (2) | 39 (3.1) | 32 (4) | **19 (4.6)** |
| enable CP | bushy | trn | 6.4e+04 (5.8e+04) | 2.4e+05 (2.4e+05) | 4.6e+04 (4.5e+04) | 3.2e+04 (2.9e+04) | 1.8e+02 (1.5e+02) | 42 (21) | 7.7e+18 (7.6e+18) | 3e+14 (3e+14) | **35 (5.5)** |
| | | val | 2e+05 (1.9e+05) | 1.1e+06 (1.1e+06) | 5.8e+04 (4.2e+04) | 6e+04 (4.7e+04) | 3.1e+02 (2.3e+02) | 1.4e+02 (78) | 4e+18 (3.9e+18) | 2.8e+14 (2.8e+14) | **1e+02 (26)** |
| | | tst | 1.6e+05 (6.9e+04) | 1.2e+05 (1.1e+05) | 2.1e+05 (1.5e+05) | 6.9e+04 (4.4e+04) | 2e+03 (1.8e+03) | 4.9e+02 (2.6e+02) | 2.2e+17 (2.2e+17) | 2.1e+17 (2.1e+17) | **2.8e+02 (43)** |
| | left | trn | 17 (8.5) | **6.3 (0.38)** | 17 (6.2) | 9.9 (3.2) | 3.6e+02 (2.9e+02) | 17 (1.4) | 7.7 (0.32) | 6.8 (0.31) | 9.9 (0.62) |
| | | val | 25 (8.9) | 15 (2.2) | 27 (7.7) | 22 (4.2) | 2e+02 (99) | 24 (2.1) | 13 (0.64) | **12 (0.57)** | 27 (2.4) |
| | | tst | 64 (21) | 36 (2.7) | 66 (16) | 59 (16) | 1.7e+03 (1.1e+03) | 1e+02 (8.7) | **28 (2.5)** | 30 (3.3) | 92 (14) |

Table 5: The results of Table 2 with standard error computed over 10 seeds.

| 95% **Quantile** | | | DQN | | DDQN | | TD3 | | SAC | | PPO |
|---|---|---|---|---|---|---|---|---|---|---|---|
| | | | RB | PER | RB | PER | RB | PER | RB | PER | |
| disable CP | bushy | trn | 80 (67) | **9.4 (0.29)** | 13 (0.68) | 14 (1.9) | 16 (1.3) | 25 (3.7) | 85 (14) | 48 (7.9) | 37 (3.5) |
| | | val | 2.6e+02 (1.1e+02) | 2.2e+02 (39) | 1.8e+02 (27) | 2e+02 (19) | 2.5e+02 (43) | **1.5e+02 (20)** | 2.3e+02 (25) | 1.8e+02 (23) | 4.1e+02 (63) |
| | | tst | 1.5e+03 (4e+02) | 1.1e+03 (2.5e+02) | 2e+03 (6.5e+02) | 1.3e+03 (3.1e+02) | **1e+03 (2.4e+02)** | 1.3e+03 (2.5e+02) | 2.4e+03 (3.5e+02) | 1.2e+03 (2.9e+02) | 3.7e+03 (1.3e+03) |
| | left | trn | 19 (3.8) | **18 (2.7)** | 28 (7.8) | 24 (8.4) | 32 (8) | 19 (2.3) | 47 (7.1) | 34 (5.3) | 38 (14) |
| | | val | 1.3e+02 (17) | 1.6e+02 (21) | 1.2e+02 (11) | 1.5e+02 (39) | 1.1e+02 (21) | **67 (7.8)** | 1.5e+02 (30) | 79 (8.6) | 1.2e+02 (34) |
| | | tst | 7e+02 (1.8e+02) | 6.2e+02 (79) | 8.3e+02 (1.4e+02) | 1.3e+03 (3.1e+02) | **5.2e+02 (72)** | 5.4e+02 (1.1e+02) | 1.7e+03 (2.6e+02) | 1e+03 (1.8e+02) | 5.5e+02 (1.3e+02) |
| enable CP | bushy | trn | 3.7e+08 (3.4e+08) | 3.8e+09 (3.8e+09) | 2.8e+07 (2.7e+07) | 2.9e+06 (1.9e+06) | 1.2e+04 (1.2e+04) | 4.2e+03 (3.8e+03) | 4.7e+22 (3.2e+22) | 7.6e+20 (7.3e+20) | **1.4e+03 (4.3e+02)** |
| | | val | 8.9e+07 (5.3e+07) | 5.8e+11 (5.8e+11) | 6.6e+08 (6.5e+08) | 2.6e+06 (1.2e+06) | 3e+04 (2.7e+04) | 3.6e+04 (3.3e+04) | 2.5e+24 (2.5e+24) | 5.1e+21 (3.1e+21) | **7.1e+03 (1.9e+03)** |
| | | tst | 1.9e+09 (1.8e+09) | 1.3e+08 (1.3e+08) | 1.3e+08 (1.1e+08) | 7.9e+06 (4.6e+06) | 6.1e+04 (4.7e+04) | 3e+04 (1.6e+04) | 3.6e+22 (3.5e+22) | 1.8e+24 (1.6e+24) | **2.1e+04 (4e+03)** |
| | left | trn | 1.4e+02 (1.1e+02) | **22 (2.2)** | 1.1e+02 (52) | 54 (30) | 5.9e+03 (4.3e+03) | 83 (17) | 28 (1.4) | 23 (1.9) | 1.6e+02 (30) |
| | | val | 2e+02 (56) | 1.9e+02 (45) | 3.9e+02 (1.5e+02) | 2.5e+02 (68) | 1.1e+04 (7.5e+03) | 2.7e+02 (20) | **76 (7)** | 82 (8.8) | 5.9e+02 (66) |
| | | tst | 2.4e+03 (7.7e+02) | 1.6e+03 (3.9e+02) | 3.8e+03 (1.3e+03) | 1.8e+03 (3.9e+02) | 4.7e+04 (2.2e+04) | 5.8e+03 (1.3e+03) | **1.1e+03 (2.4e+02)** | 1.3e+03 (2.5e+02) | 7.5e+03 (9.5e+02) |

| 99% **Quantile** | | | DQN | | DDQN | | TD3 | | SAC | | PPO |
|---|---|---|---|---|---|---|---|---|---|---|---|
| | | | RB | PER | RB | PER | RB | PER | RB | PER | |
| disable CP | bushy | trn | 1.8e+04 (1.6e+04) | **55 (3.8)** | 7e+02 (4.6e+02) | 4.2e+02 (2.2e+02) | 2.4e+03 (1.4e+03) | 3e+03 (1.5e+03) | 3.9e+04 (9e+03) | 2.2e+04 (1e+04) | 4e+04 (1e+04) |
| | | val | 5.1e+05 (2e+05) | 3.1e+05 (7.1e+04) | 2.7e+05 (7.2e+04) | 3.3e+05 (9e+04) | 2.7e+05 (7.3e+04) | **1.9e+05 (5.2e+04)** | 4.3e+05 (5.2e+04) | 4.7e+05 (6e+04) | 5.7e+05 (6.5e+04) |
| | | tst | **3.3e+05 (3.3e+04)** | 7.5e+05 (4.1e+05) | 6e+05 (1.6e+05) | 5.4e+05 (1.3e+05) | 4.5e+05 (9e+04) | 4.6e+05 (3.3e+04) | 6.2e+05 (1.1e+05) | 4e+05 (8.6e+04) | 6e+05 (5e+04) |
| | left | trn | 2e+03 (9.1e+02) | **1.5e+03 (8.3e+02)** | 7e+03 (3.6e+03) | 3.3e+03 (2.2e+03) | 9.6e+03 (4.1e+03) | 2e+03 (7.6e+02) | 9.9e+03 (3.7e+03) | 4.6e+03 (2.3e+03) | 2.8e+04 (1.6e+04) |
| | | val | 3e+05 (3.9e+04) | 2.5e+05 (2.8e+04) | 2.2e+05 (4.4e+04) | 1.4e+05 (2.4e+04) | 2e+05 (3.8e+04) | 1.5e+05 (2.6e+04) | 2.4e+05 (3.1e+04) | **1.2e+05 (2.1e+04)** | 2.1e+05 (3.3e+04) |
| | | tst | 3.1e+05 (3.2e+04) | 4.6e+05 (6.5e+04) | 4.5e+05 (5.9e+04) | 5.7e+05 (1.1e+05) | **3e+05 (3.2e+04)** | 3.3e+05 (2.4e+04) | 4.3e+05 (4.3e+04) | 3.7e+05 (3.3e+04) | 3.1e+05 (4e+04) |
| enable CP | bushy | trn | 4.3e+26 (3.7e+26) | 5.7e+23 (5.7e+23) | 1.3e+17 (1e+17) | 7.9e+14 (7.7e+14) | **4.2e+05 (1.6e+05)** | 2.5e+06 (1.6e+06) | 1.8e+37 (1.1e+37) | 3.6e+42 (3.6e+42) | 1.6e+06 (4.2e+05) |
| | | val | 1.2e+22 (1.2e+22) | 5.7e+24 (5.7e+24) | 2.8e+23 (2.1e+23) | 1.7e+14 (9.8e+13) | **2e+06 (4.4e+05)** | 3.1e+06 (1.8e+06) | 3.7e+38 (2.9e+38) | 4.2e+40 (2.8e+40) | 5.6e+06 (1.5e+06) |
| | | tst | 2.5e+29 (2.5e+29) | 5.7e+25 (5.7e+25) | 1e+23 (1e+23) | 6.5e+18 (6.4e+18) | **2.6e+06 (5.9e+05)** | 4.1e+06 (1.6e+06) | 9.2e+39 (9.2e+39) | 1.3e+41 (1.3e+41) | 7.6e+06 (2.3e+06) |
| | left | trn | 7.1e+04 (5.5e+04) | 3.5e+03 (1.2e+03) | 3.7e+04 (1.9e+04) | 1.9e+04 (1.3e+04) | 1e+06 (5.1e+05) | 7.6e+04 (2.3e+04) | 5.1e+03 (1.8e+03) | **1.9e+03 (6.9e+02)** | 4.3e+05 (1.1e+05) |
| | | val | 2.3e+05 (7.9e+04) | 2.4e+05 (4.6e+04) | 3.1e+05 (5.6e+04) | 3.6e+05 (8.6e+04) | 1.6e+06 (3.1e+05) | 2.6e+05 (4.1e+04) | **1.3e+05 (4.5e+04)** | 1.4e+05 (2.5e+04) | 6.3e+05 (1.6e+05) |
| | | tst | 1.8e+06 (7.1e+05) | 7.6e+05 (1.6e+05) | 7.9e+05 (1e+05) | 1.4e+06 (3.6e+05) | 4.8e+06 (7.6e+05) | 7.6e+06 (6.1e+06) | **3.8e+05 (4.1e+04)** | 4.4e+05 (3.6e+04) | 3.3e+06 (1.9e+06) |

Table 6: The 95% and 99% quantile of CMMs with standard error computed over 10 seeds.

### G.1 ADDITIONAL LEARNING CURVES

In this section, we plot the learning curves of the best performing algorithm in each of the four possible settings in {disable CP, enable CP} × {left, bushy}. Performance is measured by mean CCM (from Table 2) and we plot the average performance over 10 runs (shaded region is stderr over said runs). For each setting, we also show the CCM distribution over the training, validation and testing query sets in a complementary CDF (CCDF) plot. The CCDF shows that these distributions are long-tailed.

#### G.1.1 BUSHY PLANS WITH DISABLE CPS HEURISTIC

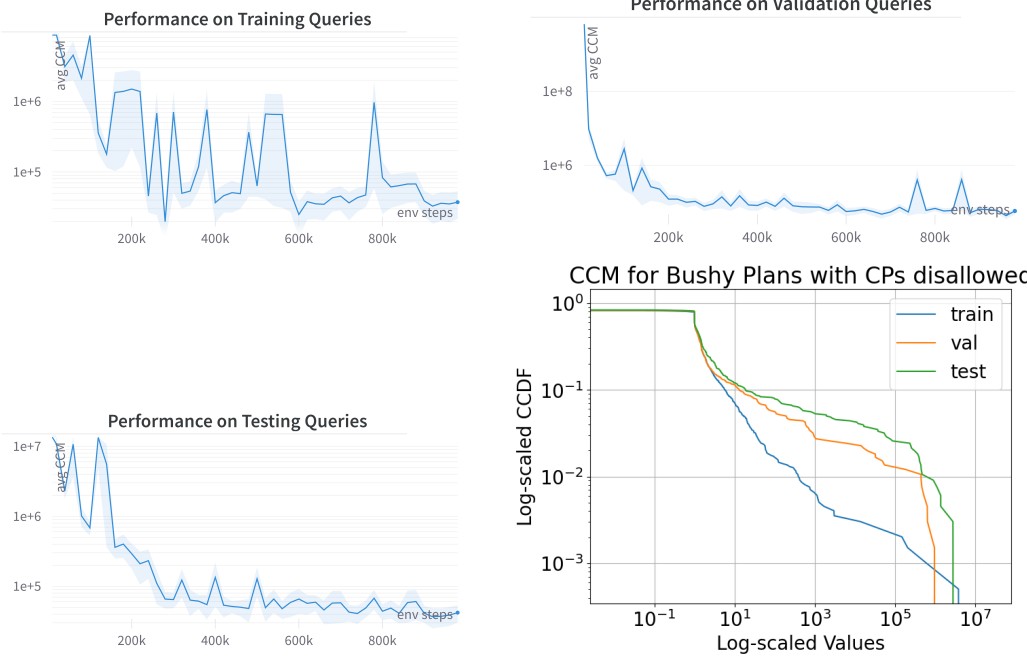

Figure 6: The best algorithm (in terms of CCM mean) for bushy plans with disable CP heuristic is TD3 with prioritized replay with policy learning rate 0.0003 and critic learning rate 0.0001. Shaded region is stderr over 10 runs.

### G.1.2 LEFT-DEEP PLANS WITH DISABLE CP HEURISTIC

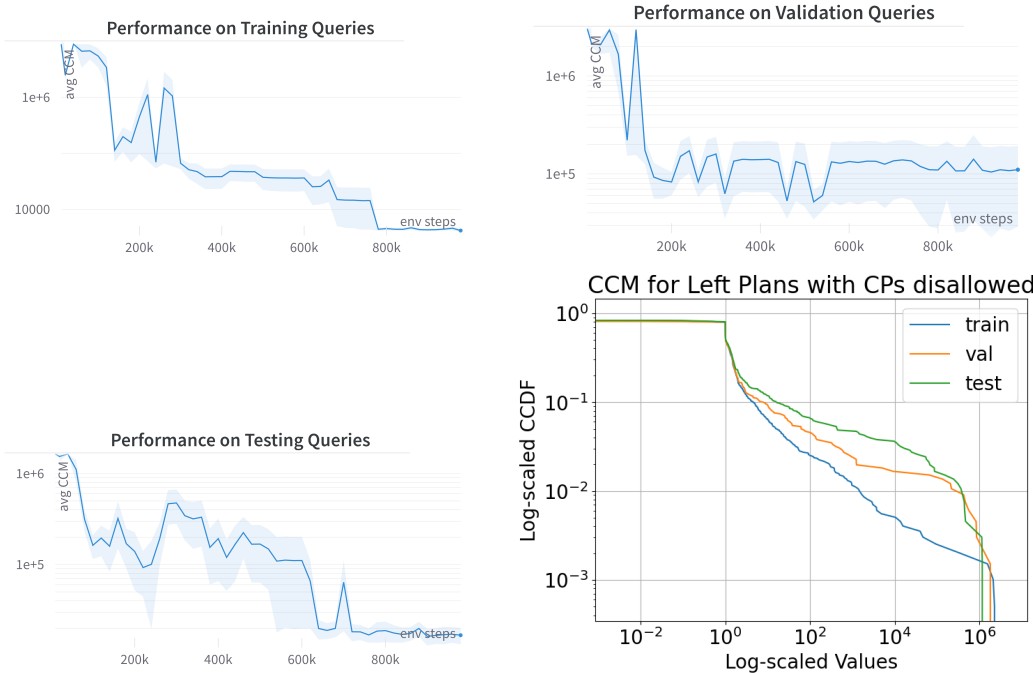

Figure 7: The best algorithm (in terms of CCM mean) for left-deep plans with disable CP heuristic is TD3 with prioritized replay with policy learning rate 0.0001 and critic learning rate 0.0003. Shaded region is stderr over 10 runs.

### G.1.3 BUSHY PLANS WITH CPs

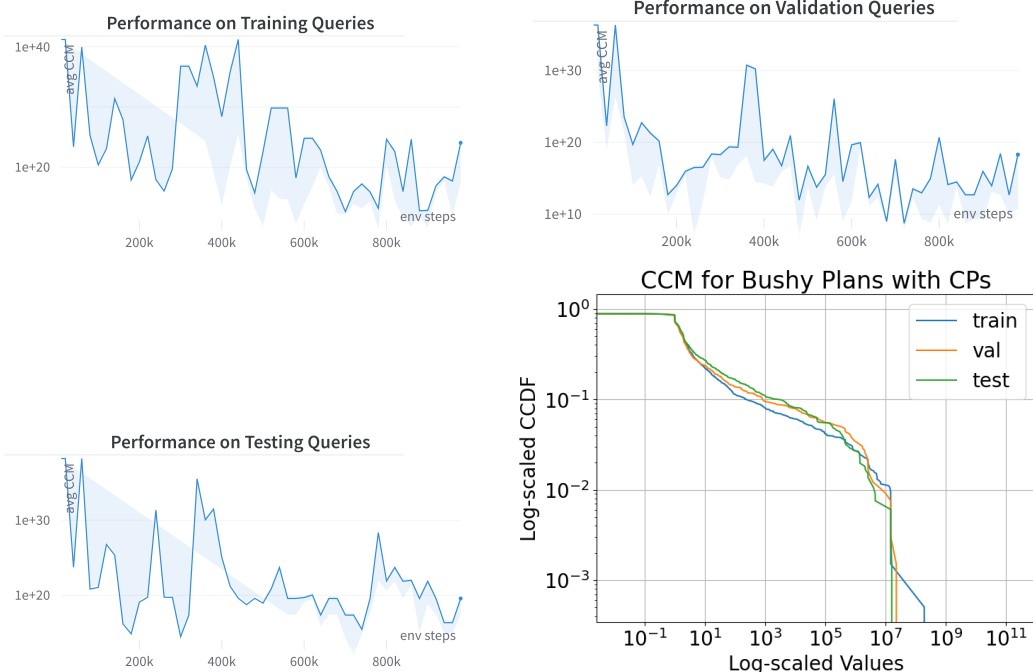

Figure 8: The best algorithm (in terms of CCM mean) for bushy plans with CPs enabled is TD3 with prioritized replay with policy learning rate 0.0003 and critic learning rate 0.0003. Shaded region is stderr over 10 runs.

### G.1.4 LEFT-DEEP PLANS WITH CPS

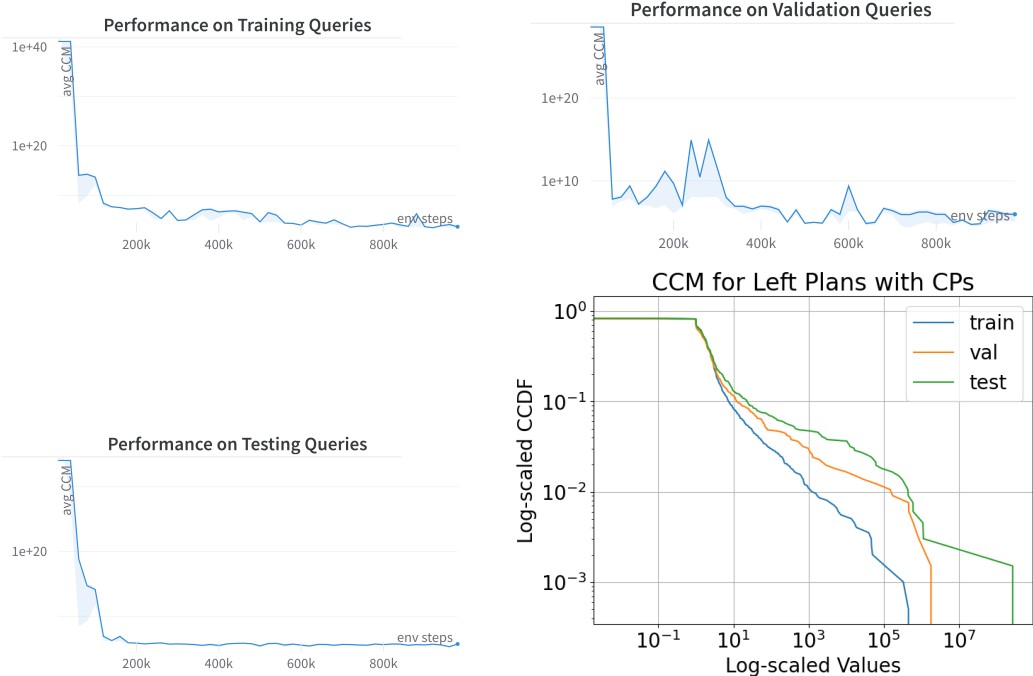

Figure 9: The best algorithm (in terms of CCM mean) for left-deep plans with CPs enabled is TD3 with policy learning rate 0.0001 and critic learning rate 0.0003. Shaded region is stderr over 10 runs.

### G.1.5   HEURISTIC METHOD

We also plot results for a heuristic `ikkbzbushy` which builds join selectivity estimates based on the training data and uses dynamic programming (DP) to compute the best bushy plan according to the estimated IR cardinalities (based on the join selectivity estimates). Similar with RL agents which learns in training queries, we build selectivity estimation of join predicates using training queries. We follow the most classic approach (Ramakrishnan & Gehrke, 2003) for estimating selectivities of each join pair $R \bowtie_P S$ using $\text{Sel}_i(R \bowtie_P S) = \frac{|R \bowtie_P S|}{|R||S|}$ and take the average among all queries as the final selectivities. And in the validate and test, we estimate the IR size using $\text{Sel}_i(R \bowtie_P S)|R||S|$. `ikkbzbushy` can guarantee that the final plan has the smallest estimation cost.

So amongst all heuristics that use the same estimated IR cardinalities, this approach is the best possible heuristic since it uses the plan with the smallest estimated cost. However, since the selectivity estimation step is biased, the final performance is very poor, and worse than the RL-based approaches. It's worth mentioning that DP-based approaches can take hours on med-large size queries since the problem space grows exponentially. In contrast, RL methods are much faster to run.

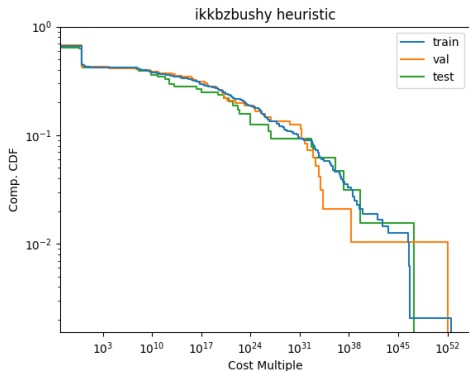

Figure 10: Distribution of cost multiples for the heuristic. The mean performance is train $7.8e + 49$, val $1.1e + 50$ and test $2.5e + 45$. Notice that these are many orders of magnitude worse than the RL policy's performance.

# H  BENCHMARKING OFFLINE RL

We now benchmark offline RL algorithms on optimizing left-deep plans with CPs disallowed. In this section, we focus on the 113 queries from the JOB (Leis et al., 2015) which is a smaller and easier setting than our main dataset of 3300 queries. We focus on the JOB for offline RL because Leis et al. (2015) provided behavior policy traces for all 113 queries, based on popular search heuristics (described below).

| JOB Data | | | DQN | | DDQN | | BC | | BCQ | | CQL | |
|---|---|---|---|---|---|---|---|---|---|---|---|---|
| | | | Median | STD | Median | STD | Median | STD | Median | STD | Median | STD |
| disable CP | left | trn | **3.22** | 1.9e6 | 1471 | 3.4e10 | 1.6e5 | 7.6e+11 | 3.1e3 | 2.7e13 | 79 | 4.1e12 |
| | | tst | **3.19** | 1.9e6 | 1470 | 3.3e10 | 8.0e4 | 7.3e12 | 9908 | 2.6e13 | 76 | 3.9e12 |
| | | val | **3.21** | 2.0e6 | 1.0e3 | 3.4e10 | 6.4e5 | 7.5e12 | 1.5e5 | 2.9e13 | 81 | 4.1e12 |

Table 7: CCM (lower is better) averaged over the training (trn), validation (val) or testing (tst) query sets.

**Experimental Setup**  Our offline dataset is comprised of trajectories from the following behavior policies provided by the JOB (Leis et al., 2015): adaptive, dphyp, genetic, goo, goodp, goodp2, gooikkbz, ikkbz, ikkbzbushy, minsel, quickpick, simplification (Neumann & Radke, 2018). We highlight that these behavior policies are *search* heuristics, which operate given a cost model, *e.g.*, estimated IR cardinalities, to plan over. To generate behavior trajectories, we provided these heuristics access to the ground-truth IR cardinalities. Alternatively, one could take traces from existing DBMS such as PostgreSQL. For each heuristic, we collected 1000 trajectories across different queries. We partition the dataset for training, evaluation and testing similarly as in our online experiments.

**Discussions**  Table 7 summarizes our offline results. We find that DQN has the best performance in terms of median and the validation/testing results are even better than online. CQL also obtains reasonable performance, but all other methods seem to have relatively poor median performance even on the training set. It's worth noting that all methods seem to have a heavy tail performance distribution (over queries), as shown by the large standard deviations. In later sections of the appendix, we see this is the case for online RL as well. This heavy-tail distribution of returns motivates applying risk-sensitive RL methods to JOINGYM for future work.

We also tested on some other offline algorithms, such as SAC, and it is hard to converge hence we didn't report the results. We observe that the TD error is increasing, although the Q value functions, actors and critics are learning. Making too many TD updates to the Q-function in offline deep RL is known to sometimes lead to performance degradation and unlearning, we can use regularization to address the issue (Kumar et al., 2021).

## H.1  HYPERPARAMETERS FOR OFFLINE RL ALGORITHMS

We performed hyperparameter search with grid search and Bayesian optimization. The final parameters we used for evaluation is shown below in Tables 9-12.

### H.1.1  BATCH-CONSTRAINED Q-LEARNING

Table 8: Hyperparameter of Batch-Constrained Q-learning algorithm (BCQ).

| Hyperparameter | Value |
|---|---|
| Learning rate | $6.25 \times 10^{-5}$ |
| Optimizer | Adam ($\beta = (0.95, 0.999)$) |
| Batch size | 32 |
| Number of critics | 6 |
| Discount factor | 0.99 |
| Target network synchronization coefficiency | 0.005 |
| Action flexibility | 0.3 |
| Gamma | 0.99 |

### H.1.2 BEHAVIOR CLONING

Table 9: Hyperparameter of Behavior Cloning (BC).

| Hyperparameter | Value |
|---|---|
| Learning rate | 0.001 |
| Optimizer | Adam ($\beta = (0.9, 0.999)$) |
| Batch size | 100 |
| Beta | 0.5 |

### H.1.3 CONSERVATIVE Q-LEARNING

Table 10: Hyperparameter of Conservative Q-Learning (CQL).

| Hyperparameter | Value |
|---|---|
| Actor learning rate | $3 \times 10^{-4}$ |
| Critic learning rate | $3 \times 10^{-4}$ |
| Learning rate for temperature parameter of SAC | $1 \times 10^{-4}$ |
| Learning rate for alpha | $1 \times 10^{-4}$ |
| Batch size | 256 |
| N-step TD calculation | 1 |
| Discount factor | 0.99 |
| Target network synchronization coefficiency | 0.005 |
| The number of Q functions for ensemble | 2 |
| Initial temperature value | 1.0 |
| Initial alpha value | 1.0 |
| Threshold value | 10.0 |
| Constant weight to scale conservative loss | 5.0 |
| The number of sampled actions to compute | 10 |

### H.1.4 DQN

Table 11: Hyperparameter of DQN.

| Hyperparameter | Value |
|---|---|
| Learning rate | 6.25e-4 |
| Batch size | 32 |
| target_update_interval | 8000 |

### H.1.5 DOUBLE DQN

Table 12: Hyperparameter of DDQN.

| Hyperparameter | Value |
|---|---|
| Learning rate | 6.25e-4 |
| Batch size | 32 |
| target_update_interval | 8000 |

