# OpenReview forum: "JoinGym: An Efficient Query Optimization Environment for Reinforcement Learning"
_ICLR.cc/2024/Conference — Submitted to ICLR 2024_

### Official Review · Reviewer_jBfS · 2023-10-29

**Soundness:** 3 good
**Presentation:** 2 fair
**Contribution:** 2 fair
**Rating:** 5
**Confidence:** 3

**Summary:**

The problem of "join plan enumeration" is addressed using the Partially Observable Contextual Markov Decision Process (POCMDP). The authors conducted a comparative evaluation of join order optimization methods that utilize reinforcement learning, using a set of join query plans generated with the proposed technique.

**Strengths:**

S1: This is a new benchmark paper for various join order optimization methods that utilize reinforcement learning (RL).

S2: The research on join query plan enumeration itself is novel. Additionally, it is valuable for estimating the cost (CCM) for each query plan.

**Weaknesses:**

W1: The paper will become more valuable by additionally discussing the pros and cons of the compared reinforcement learning (RL) methods and clearly outlining future research challenges.

W2: The proposal enumerates join query plans with its cost model (CCM). Since the proposal utilizes lossy table embedding, it's important to compare the accuracy of the CCM to existing methods, such as Neo (Marcus et al. 2019).

W3: Most parts of the paper heavily rely on knowledge of Markov Decision Process (MDP), which makes it difficult to read. I suggest the authors provide a preliminary overview of MDP to improve readability.

W4: The abstract seems somewhat inconsistent with the main content of the paper. For example, the authors state "key advantages of JOINGYM are usability and significantly higher throughput," but there is no description in the main content that pertains to usability and throughput.

**Questions:**

Q1. How do you estimate c_h (the cardinality of the IR incurred at time h)?

---

> ### Author Response · Authors · 2023-11-16
>
> Thank you for your valuable feedback and please find our responses below.
>
> **W1.** We've outlined three concrete future research directions as 1) long-tailed returns, 2) generalization in combinatorial optimization and 3) partial observability. Admittedly, our discussion section was cut short due to space constraints. Hence, we will add the following paragraph discussing the pros and cons of compared RL methods:
>
> To recap, we compared four RL algorithms spanning three categories: off-policy Q-learning (DQN), off-policy actor-critic (TD3 and SAC), and on-policy actor-critic (PPO). Although PPO does not always obtain the best test-set performance, we can see in Tables 5,6 that it reliably obtains decent quantile performance in all settings of JoinGym. Hence, PPO has the advantage of being the most stable and the disadvantage of often plateauing at a sub-optimal solution. On the other hand, off-policy actor-critic methods (TD3 and SAC) perform the best for mean and $99\%$ quantile, although PPO's CCM is usually within an order of magnitude. Hence, off-policy actor-critic methods are less stable but can sometimes converge to a better solution. DQN and DDQN tend to be uniformly worse off, which suggests that actor critic outperforms Q-learning for this problem. With this said, all of these algorithms are optimizing for the mean of a highly varied, long-tailed distribution of returns; thus, designing better algorithms that can handle long-tailed rewards will likely improve performance across all the algorithms.
>
> **W2 and Q1:** We never estimate cardinalities since JoinGym can simulate *true* cardinalities derived from our novel dataset. Thus, it does not make sense to compare against existing cost models since JoinGym does not require a cost model. Again, this is a key advantage of JoinGym.
>
> **W3.** Please see Section 4.1 for the MDP setup. We kept it short due to space constraints and can certainly point the reader to a longer exposition, e.g. Sutton and Barto, 2018.
>
> **W4.** Please see our discussion on usability and higher throughput in "Related works: Environments for Query Optimization" and a longer discussion in Appendix C. We acknowledge that this could have been highlighted better and we will amend our paper with the following paragraph:
>
> JoinGym improves over prior query optimization simulators in two main ways: easier to use and higher throughput. First, JoinGym is easier to use because we do not require the user to set up a DBMS or cost model while prior simulators do. In other words, starting up JoinGym is as simple as starting up CartPole or Mountain Car in OpenAI Gym. Second, JoinGym can achieve a much higher throughput because each simulator step is simply a table lookup from our novel dataset. Prior simulators either execute the query (which can take hours to days for large queries) or estimate the query cost with a cost model (which can still take seconds), both of which are slower and require more infrastructure setup. Indeed, JoinGym can reliably simulate thousands of queries per second on a standard laptop. Moreover, the cost model can have estimation errors, while JoinGym always returns the true cardinality.
>
> Thank you for your valuable feedback and please let us know if you have additional questions.

---

> > ### Comment · Reviewer_jBfS · 2023-11-23
> >
> > Thanks for the response. According to the response, I raised the score to 5: marginally below the acceptance threshold.
> >
> > Allow me to provide further clarification (apologies for the delayed response).
> > 1) Effectiveness of busy plans: Most JOS techniques predominantly focus on left-deep joins, assuming that the search space for bushy plans is extensive and that most of these plans are ineffective. However, it appears the authors regard certain bushy plans as potentially useful (possibly being the best plans), given that experiments were conducted on both left-deep and bushy plans. My question is: what advantage is gained by expanding the search space from solely left-deep plans to include bushy plans? If a substantial number of bushy plans prove to be superior, then broadening the search space to incorporate them would seem reasonable.
> > 2) The cost of RL: RL incurs significant costs, presenting a trade-off between efficiency and quality. Given JOINGYM's focus on benchmarking RL-based techniques for JOS, could you provide specific practical scenarios where RL-based methods outperform other techniques?
> > 3) It might be beneficial if the authors referenced recent non-RL works in JOS, like "Efficiently Computing Join Orders with Heuristic Search," presented at SIGMOD2023. This work demonstrated that "heuristic search finds optimal plans an order of magnitude faster than the current state of the art."

---

> ### Author Response · Authors · 2023-11-23
>
> Dear Reviewer, thanks very much for replying during the discussion period and raising more valuable questions. Please find our responses below.
>
> **1.** JoinGym includes both left-deep and bushy plans so that it can faithfully simulate different modern DB systems. For example, Oracle only considers left-deep plans, while Postgres also considers bushy plans. Leis et al. (2015) also demonstrate that bushy plans can result in lower costs than the optimal left-deep plans. However, in our experiments, we found that bushy plans are much harder to learn with RL and almost always have worse performance than left-deep policies, likely due to the much larger search space as you pointed out. Nonetheless, we include bushy plan as an option in JoinGym to provide a realistic simulation environment for RL researchers.
>
> **2.** RL approaches can indeed incur costs during training, but the learned RL policies can often suggest better join plans than non-learning approaches at inference time. That is, RL is potentially better at inference while incurring costs at training. In particular, we have **new results** on how well Postgres, a strong non-learning baseline, performs on the JoinGym query suite. We ran the Postgres query optimizer on all 3300 queries of JoinGym and computed the join plans' cumulative cost multiples (CCMs). The results are shown below:
> | Query Set | Mean | p90 | p95 | p99 |
> | --------- | ---- | --- | --- | --- |
> | Train     | 1.72e+06 | 2.60e+03 | 4.92e+04 | 4.52e+06 |
> | Val       | 5.81e+05 | 4.46e+03 | 9.97e+04 | 7.10e+06 |
> | Test      | 6.09e+04 | 5.00e+03 | 3.41e+04 | 2.31e+06 |
>
> By examining Tables 5 and 6 of the paper, we can see that RL algorithms in the left-deep disable CP JoinGym often yields better results than Postgres. For example, PPO is consistently better for train, validation and test queries. This suggests that these RL algorithms are indeed proposing high-quality joins that are competitive with Postgres.
>
> **3.** Thanks for pointing out (Haffner and Dittrich, 2023), which we'll definitely discuss and cite. (Haffner and Dittrich, 2023) focuses on improving heuristic search algorithms for shaped queries (e.g., star and clique queries), while our paper focuses on providing a valuable realistic simulator for the RL community. We highlight that our work is positioned as a "dataset and benchmark" contribution (as marked by the primary area of our submission), where our focus is *not* to directly improve state-of-the-art in DB systems. Rather, our key contribution is a new efficient RL environment that enables anyone with a laptop to rapidly prototype new RL ideas for the query optimization problem, without needing expensive and complex setup with real DB systems. Please also see our global response regarding this point.
>
> Thanks again for your valuable feedback and please let us know if you have additional questions. If you are reassured of the value of our work by the rebuttal & discussion, then we’d greatly appreciate if you could please correspondingly increase your scores. Thank you!

---

### Official Review · Reviewer_2MKf · 2023-11-01

**Soundness:** 3 good
**Presentation:** 2 fair
**Contribution:** 3 good
**Rating:** 6
**Confidence:** 4

**Summary:**

The paper proposes a simulator for training RL agents to perform the task of DB query optimization. The main issue that the contribution tackles is the cost of running expensive queries on real hardware which can hinder learning efficiency (from a wall-time perspective) when training deep reinforcement learning agents. This becomes even more problematic in the case of query optimization where the search space is exponential and finding fast-executing query plans is NP-hard. The simulator uses precomputed exact cardinality for each of the possible plans in the search space, which offloads the cost of evaluating plan performance (and therefore collecting a reward) to the developers of the simulator rather than the user. The authors release the queries and intermediate relation cardinality estimates which can also be used for learning cardinality estimation models. Many RL algorithms are benchmarked using this RL environment.

**Strengths:**

Many reinforcement learning algorithms are benchmarked against the proposed simulator and the release of the dataset along with the precomputed IR cardinalities will be of use to the database and QO research communities. The background on database query optimization and traditional QO optimizations (e.g. left-deep vs bushy) also helps contextualize the contribution to the broader machine learning community and is presented well.

**Weaknesses:**

With respect to the analysis of the environment and the cost models associated, the paper does not benchmark against performance of existing RL-for-QO systems, such as Neo (Marcus et. al) which do indeed use expensive execution frameworks. Additionally, more traditional cost models which approximate the IR estimates such as the Postgres cost model should be evaluated to solidify that precomputing the IR estimates manually is necessary for strong performance. Ideally, showing a curve of environment cost vs. learning performance curve would be useful (e.g. does using real execution latencies improve over the exact IR cardinalities which improves over the Postgres cost model?).

The dataset for learning supervised cardinality estimators is not benchmarked against other methods for learning cardinality estimators (both supervised and unsupervised) and should be benchmarked to better contextualize the contribution for the broader ML and systems community as to the datasets potential impact with respect to training estimators.

**Questions:**

Installing postgres and querying the cost model is not particularly expensive or difficult and could be packaged into software such as an OpenAI gym environment. How does the performance compare when using the postgres cost model? What about using more advanced cardinality estimators, such as NeuroCard (Yang et al) as the cost model for the JoinGym simulator?

What is the performance on JOB-Ext (Marcus et al, available https://github.com/RyanMarcus/imdb_pg_dataset/tree/master/job_extended) which are out-of-domain queries and not from the original JOB templates?

How does the method generalize to new schemas?

---

> ### Author Response · Authors · 2023-11-20
>
> Thank you for your valuable feedback and please find our responses below.
>
> **W1a.** As you said, Neo (Marcus et al., 2019) indeed uses expensive execution frameworks, whereas our work focuses on lightweight simulation and so our benchmark results are not directly comparable with Neo's. Our emphasis is to provide a lightweight simulator for the RL community rather than to directly improve DB systems. Please also see our global response regarding this point.
>
> **W1b.** Thanks for suggesting Postgres. We have new results on how well Postgres, a strong non-learning baseline, performs on the JoinGym query suite. We ran the Postgres query optimizer on all 3300 queries of JoinGym and computed the join plans' cumulative cost multiples (CCMs). The results are shown below:
> | Query Set | Mean | p90 | p95 | p99 |
> | --------- | ---- | --- | --- | --- |
> | Train     | 1.72e+06 | 2.60e+03 | 4.92e+04 | 4.52e+06 |
> | Val       | 5.81e+05 | 4.46e+03 | 9.97e+04 | 7.10e+06 |
> | Test      | 6.09e+04 | 5.00e+03 | 3.41e+04 | 2.31e+06 |
>
> By examining Tables 5 and 6 of the paper, we can see that RL algorithms in the left-deep disable CP JoinGym often yields better results than Postgres. For example, PPO is consistently better for train, validation and test queries. This suggests that these RL algorithms are indeed proposing high-quality joins that are competitive with Postgres.
>
> **W2.** Our work focuses on providing a RL environment for query optimization. We believe our dataset can be useful for cardinality estimation research, but since our focus is RL, we leave supervised learning benchmarks as promising future work.
>
> **Q1a.** Please see **W1b** for our new results on Postgres's performance on JoinGym.
>
> **Q1b.** Thanks for suggesting Neurocard, which we will definitely discuss and cite. Since JoinGym already returns true cardinalities, using NeuroCard as the cost model for JoinGym does not make sense because it would only introduce estimation error and slow-down the simulation throughput.
>
> **Q2.** Our experiments are already run on a large dataset of $3300$ queries with $660$ as the test set. JOB-Ext consists of $24$ queries and is not a standard benchmark for query optimization, so adding these queries would not add much to JoinGym.
>
> **Q3.** This is a great question. If you are wondering whether it is possible to use JoinGym to simulate another schema or database, this is definitely possible -- all one needs to do is to collect a new cardinality dataset which is straightforward but computationally expensive. However, if you are wondering whether it is possible to transfer an RL policy trained on IMDb to another schema or database, this is a much harder problem. Even with same schema and fixed database, our experiments show that there is much to be improved along the three directions of (1) long-tailed returns, (2) generalization in discrete problems, and (3) partial observability. Extending JoinGym to transfer between schemas is certainly promising future work.
>
> Thanks again for your valuable feedback and please let us know if you have additional questions.

---

> > ### Comment · Reviewer_2MKf · 2023-11-22
> >
> > Thank you for the additional experiments and for the thorough response. The positioning of this work as a novel environment for RL research makes sense and the additional Postgres ablation is helpful for understanding the method. I have adjusted my score accordingly.

---

> > > ### Author Response · Authors · 2023-11-23
> > >
> > > Thank you so much for replying and increasing your score. We’re grateful for your valuable feedback and will integrate them into our paper.

---

### Official Review · Reviewer_QAMY · 2023-11-01

**Soundness:** 3 good
**Presentation:** 3 good
**Contribution:** 3 good
**Rating:** 5
**Confidence:** 2

**Summary:**

The work is related to applying reinforcement learning to decide how a join query (a query that cross-references multiple tables) can be executed most efficiently by a database system which is an NP-hard problem where existing solutions still leave a lot of room for improvement. The paper proposes a simulator/benchmark to evaluate the performance of reinforcement learning approaches and features an extensive empirical study that compares some existing approaches.

**Strengths:**

S1) Originality: Thorough investigation of existing reinforcement learning approaches and useful tool/benchmark for future works.

S2) Significance: Reinforcement learning is a reasonable approach for the join order problem and shedding light on what works best can contribute towards faster join execution not just in general, but also for particular data domains / systems.

S3) Presentation: The paper presents ideas clearly, particularly figures seem to illuminate concepts well and the literature review appears to be thorough.

**Weaknesses:**

W1) Significance: The conclusion of compared reinforcement learning approaches could be clearer in terms of which kind of predictions it enables. A non-learning baseline would also help to root the results.

W2) Originality: The work does not seem to (explicitly) propose any novel approach.

W3) Relevance: The focus of the paper seems to be less on reinforcement learning and more about a particular application of reinforcement learning in database management systems.

**Questions:**

Q1) What can be learned about the performance of different reinforcement learning approaches in this benchmark and how does it compare to the performance of a non-learning approach (as used in modern commercial systems)?

Q2) Do the any of the results conclude any new approach or variation of an approach that has not been previously considered?

Q3) What kind of progress does this work make in terms of reinforcement learning approaches (e.g., via trial and error) beyond measuring their empirical performance?

---

> ### Author Response · Authors · 2023-11-20
>
> Thank you for your valuable feedback and please find our responses below.
>
> **W1 and Q1.** Regarding what can be learned about RL approaches in the benchmark, we mentioned the most salient points in the discussion part of Section 5. To add more details, we'll amend it with the following paragraph, comparing pros and cons of compared RL methods for the query optimization task:
>
> To recap, we compared four RL algorithms spanning three categories: off-policy Q-learning (DQN), off-policy actor-critic (TD3 and SAC), and on-policy actor-critic (PPO). Although PPO does not always obtain the best test-set performance, we can see in Tables 5,6 that it reliably obtains decent quantile performance in all settings of JoinGym. Hence, PPO has the advantage of being the most stable and the disadvantage of often plateauing at a sub-optimal solution. On the other hand, off-policy actor-critic methods (TD3 and SAC) perform the best for mean and
>  quantile, although PPO's CCM is usually within an order of magnitude. Hence, off-policy actor-critic methods are less stable but can sometimes converge to a better solution. DQN and DDQN tend to be uniformly worse off, which suggests that actor critic outperforms Q-learning for this problem. With this said, all of these algorithms are optimizing for the mean of a highly varied, long-tailed distribution of returns; thus, designing better algorithms that can handle long-tailed rewards will likely improve performance across all the algorithms.
>
> Regarding comparison with a non-learning approach: we have new results on how well Postgres, a strong non-learning baseline, performs on the JoinGym query suite. We ran the Postgres query optimizer on all 3300 queries of JoinGym and computed the join plans' cumulative cost multiples (CCMs). The results are shown below:
> | Query Set | Mean | p90 | p95 | p99 |
> | --------- | ---- | --- | --- | --- |
> | Train     | 1.72e+06 | 2.60e+03 | 4.92e+04 | 4.52e+06
> | Val       | 5.81e+05 | 4.46e+03 | 9.97e+04 | 7.10e+06
> | Test      | 6.09e+04 | 5.00e+03 | 3.41e+04 | 2.31e+06
>
> By examining Tables 5 and 6 of the paper, we can see that RL algorithms in the left-deep disable CP JoinGym often yields better results than Postgres. For example, PPO is consistently better for train, validation and test queries. This suggests that these RL algorithms are indeed proposing high-quality joins that are competitive with Postgres.
>
> **W2 and Q2.** Our contribution is a novel RL environment JoinGym and extensive benchmarking of existing RL algorithms. As a submission for the "dataset and benchmarks" primary area, our goal is to provide a RL environment to accelerate and facilitate future algorithm research.
>
> **W3 and Q3.** We believe our work will be valuable to the RL community for the following reasons. First, as mentioned in the previous point, our new efficient environment will enable anyone with a laptop to rapidly prototype new RL ideas for the query optimization problem, without needing expensive and complex setup with real DB systems. Second, in Section 5 of the paper, we have identified three clear research directions for RL: (1) dealing with long-tailed returns, (2) generalization in discrete optimization, and (3) partial observability. Please also see our global response regarding this point.
>
> Thanks again for your valuable feedback and please let us know if you have additional questions.

---

> > ### Comment · Reviewer_QAMY · 2023-11-22
> >
> > Thank you very much that clears up the questions I had! Postgres apparently does an exhaustive search for smaller problem instances and genetic algorithms for larger join queries (https://www.postgresql.org/docs/current/geqo-pg-intro.html), which confirms that the optimisation problem is quite hard (making it an interesting target to learn more about RL) and could benefit from fresh perspectives (making RL an interesting approach for the problem).

---

> > > ### Author Response · Authors · 2023-11-22
> > >
> > > Dear Reviewer, thanks so much for replying! As you point out, join order selection is indeed a difficult optimisation problem and can benefit from learning-based approaches which our simulator can accelerate the research of. Please let us know if there are remaining or new questions you would like us to address. If you are reassured of the value of our work by the rebuttal & discussion, then we’d greatly appreciate if you could please correspondingly update your scores. Thank you!

---

### Official Review · Reviewer_wp5V · 2023-11-03

**Soundness:** 1 poor
**Presentation:** 2 fair
**Contribution:** 2 fair
**Rating:** 3
**Confidence:** 5

**Summary:**

This paper proposes a lightweight query optimization environment for reinforcement learning and releases a cardinality dataset based on IMDB.

**Strengths:**

1.This paper focus on an important problem for query optimization.

**Weaknesses:**

1. The title of the article is very strange and difficult to understand. The ‘query optimization’ is the background of this paper. The title may be “An efficient reinforcement learning environment for query optimization” instead of “An efficient query optimization environment for reinforcement learning”?
2. Do the ‘simulator’ and ‘environment’ refer to the same thing? Authors mention that ‘our aim is to provide a lightweight yet realistic simulator’ and ‘looking up the size of intermediate tables from join sequences’. The size or called cardinality estimation is a popular research filed in query optimization. They should compare their method to the existing query size simulators as follows.
[1] Sun, J., & Li, G. (n.d.). An End-to-End Learning-based Cost Estimator. VLDB, 2020. Kipf [2] A, Kipf T, Radke B, et al. Learned cardinalities: Estimating correlated joins with deep learning. CIDR, 2019.
[3] Yang, Z., Kamsetty, A., Luan, S., Liang, E., Duan, Y., Chen, X., & Stoica, I. (2020). Neurocard: One cardinality estimator for all tables. Proceedings of the VLDB Endowment, 14(1), 61–73, 2020.
[4] Ziniu Wu, Parimarjan Negi, Mohammad Alizadeh, Tim Kraska, Samuel Madden. FactorJoin: A New Cardinality Estimation Framework for Join Queries. SIGMOD, 2023
3. Many statements of this paper are incorrect. For example, authors mention ‘runtime metrics are system-dependent and can only be obtained from live query executions’. In fact, there exists many popular works to simulate the query size and cost [1,2,3,4].
4. This article needs to be greatly improved in algorithm, experiment and writing before it can be accepted.

**Questions:**

Shown in weaknesses.

---

> ### Author Response · Authors · 2023-11-16
>
> Thank you for your valuable feedback and please find our responses below.
>
> **Q1.** Our title highlights that JoinGym is designed for the RL community, rather than to improve query optimization directly. Up until now, query optimization has not been easy for RL researchers to benchmark due to large overhead in setting up a real DBMS. Our work removes this barrier.  With JoinGym, we hope to motivate research from diverse ML/RL communities to create better algorithms for query optimization. Please also see our global response regarding this point.
>
> **Q2.** Yes, "simulator" is the RL environment. Thanks for bringing up prior works on cardinality estimation, which we will cite and discuss -- our work is orthogonal to [1-4] because we are using *true cardinalities* instead of estimates from cost models (which can have errors). Moreover, [1-4] do not provide a RL environment for query optimization, which is our focus and main contribution.
>
> **Q3.** By runtime metrics, we meant actual latency and wall-clock runtime of joins, which are system-dependent. Our point here is that cardinality does not depend on which machine or system the query is run on, which is one benefit to using cardinality as the cost.
>
> **Q4.** This work is positioned as a "dataset and benchmark" contribution, which we've indicated by selecting this primary area. Our goal with JoinGym is not to create new algorithms but rather to provide a valuable simulator for the RL community.  We emphasize that our work provides a novel and valuable "dataset and benchmark" for the RL community.
>
> Thanks again for your valuable feedback and please let us know if you have additional questions.

---

> > ### Author Response · Authors · 2023-11-23
> >
> > Dear Reviewer, as the open discussion period is nearing the end, please let us know if there are remaining or new questions you would like us to address. We really appreciate your time and valuable feedbacks. Thank you.

---

### Author Response · Authors · 2023-11-16
**Global Response**

Dear Reviewers,

We are grateful for your valuable reviews. Many questions were about whether our paper contains better algorithms that can directly improve DB systems. This is not our focus. This work is a "dataset and benchmark" contribution, which we've indicated by selecting this primary area. Our goal is to provide an efficient query optimization simulator to motivate research in RL algorithms for discrete optimization (see goal 1. of [John Langford's blog on RL simulators](https://hunch.net/?p=8825714)). We believe this will be useful to the RL community since join order selection has key challenges that aren't captured by typical RL environments (Atari, Procgen, MuJoCo), including long-tailed returns, generalization in combinatorial optimization and partial observability. Thus, our goal is to create a realistic and efficient simulator that can facilitate RL research for this problem. The most related work to ours is Park ([Mao et al., 2019](https://proceedings.neurips.cc/paper/2019/hash/f69e505b08403ad2298b9f262659929a-Abstract.html)), which JoinGym improves by providing much higher throughput and better usability (no need to setup a DBMS or cost model). In short, our work is a "dataset and benchmark" contribution: we provide a realistic, efficient and easy-to-use simulation environment to bridge the ML/RL and DB systems communities.

---

### Meta-Review · Area_Chair_f1k2 · 2023-12-23

**Metareview:**

The paper tackles an important problem in query optimization, providing a thorough investigation of existing reinforcement learning approaches and a valuable benchmark for future work in this area. However, It also has several key shortcomings.  The novelty is not strong; it fails to propose significant new approaches. Methodologically, it does not sufficiently compare its techniques with existing ones, weakening its empirical validity, particularly against non-learning baselines.   The readability should also be improved.  From these weak points, The present paper unfortunately does not justify acceptance for neurips.

**Justification For Why Not Higher Score:**

The novelty and comparison of the method is insufficient

**Justification For Why Not Lower Score:**

The paper provides a thorough investigation of existing RL and valuable benchmarks.

---

### Decision · Program_Chairs · 2024-01-16

Reject